# R-EDL: Relaxing Nonessential Settings of Evidential Deep Learning

**Mengyuan Chen**[1,2]**, Junyu Gao**[1,2]**, Changsheng Xu**[1,2,3*]
[1]State Key Laboratory of Multimodal Artificial Intelligence Systems (MAIS),
Institute of Automation, Chinese Academy of Sciences (CASIA)
[2]School of Artificial Intelligence, University of Chinese Academy of Sciences (UCAS)
[3]Peng Cheng Laboratory, ShenZhen, China
`chenmengyuan2021@ia.ac.cn; {junyu.gao,csxu}@nlpr.ia.ac.cn`

## Abstract

A newly-arising uncertainty estimation method named Evidential Deep Learning (EDL), which can obtain reliable predictive uncertainty in a single forward pass, has garnered increasing interest. Guided by the subjective logic theory, EDL obtains Dirichlet concentration parameters from deep neural networks, thus constructing a Dirichlet probability density function (PDF) to model the distribution of class probabilities. Despite its great success, we argue that EDL keeps nonessential settings in both stages of model construction and optimization. In constructing the Dirichlet PDF, a commonly ignored prior weight parameter governs the balance between leveraging the proportion of evidence and its magnitude in deriving predictive scores. In model optimization, a variance-minimized regularization term adopted by traditional EDL encourages the Dirichlet PDF to approach a Dirac delta function, potentially exacerbating overconfidence. Therefore, we propose the R-EDL (**R**elaxed-**EDL**) method by relaxing these nonessential settings. Specifically, R-EDL treats the prior weight as an adjustable hyperparameter instead of a fixed scalar, and directly optimizes the expectation of the Dirichlet PDF provided to deprecate the variance-minimized regularization term. Extensive experiments and SOTA performances demonstrate the effectiveness of our method. Source codes are provided in Appendix E.

## 1 Introduction

In high-risk domains such as autonomous driving and medical analysis, it is imperative for models to reliably convey the confidence level of their predictions (Choi et al., 2019; Abdar et al., 2022). However, previous research suggests that most modern deep neural networks (DNNs), especially when trained for classification via supervised learning, exhibit poor calibration, tending predominantly towards over-confidence (Guo et al., 2017). Despite effective uncertainty methods based on Bayesian theory and ensemble techniques have been developed, these mainstream methods of uncertainty quantification necessitate multiple forward passes in the inference phase (Blundell et al., 2015; Dusenberry et al., 2020; Gal & Ghahramani, 2016; Lakshminarayanan et al., 2017; Wen et al., 2020; Egele et al., 2022), imposing substantial computational burdens that hamper their widespread industrial adoption. This limitation drives the interest of researchers in exploring how to achieve high-quality uncertainty estimation with minimal additional cost.

Evidential deep learning (EDL) (Sensoy et al., 2018) is such a newly arising single-forward-pass uncertainty estimation method, which has attracted increasing attention for its success in various pattern recognition tasks (Amini et al., 2020; Bao et al., 2021; Qin et al., 2022; Chen et al., 2022; Oh & Shin, 2022; Sun et al., 2022; Park et al., 2022; Sapkota & Yu, 2022; Gao et al., 2023a). Drawing upon the theory of subjective logic (Jøsang, 2001; 2016), EDL harnesses both the proportion of collected evidence among classes and their magnitude value to achieve high-quality uncertainty estimation, effectively mitigating model over-confidence on misclassified and out-of-distribution samples. Specifically, in a $C$-class classification task, EDL constructs a Dirichlet distribution $\text{Dir}(\boldsymbol{p}_X, \boldsymbol{\alpha}_X)$

---

*Corresponding author

to model the distribution of class probability $\boldsymbol{p}_X$ under the guidance of subjective logic, and the concentration parameter vector $\boldsymbol{\alpha}_X(x)$ is given by

$$\boldsymbol{\alpha}_X(x) = \boldsymbol{e}_X(x) + C \cdot \boldsymbol{a}_X(x), \quad \forall x \in \mathbb{X} = \{1, 2, ..., C\}, \tag{1}$$

where the base rate $\boldsymbol{a}_X$ is typically set as a uniform distribution over $\mathbb{X}$, and its scalar coefficient $C$ serves as a parameter termed as a *prior weight*. Note that to keep the notation uncluttered, we use $\boldsymbol{\alpha}_X(x)$ as a simplified expression of $\boldsymbol{\alpha}_X(X = x)$, and similarly for $\boldsymbol{e}_X(x)$ and $\boldsymbol{a}_X(x)$. The random variable $X$ denotes the class index of the input sample, and $\boldsymbol{e}_X(x)$ signifies the amassed evidence for the sample's association with class $x$. Thereafter, for model optimization, the traditional EDL method integrates the mean square error (MSE) loss over the class probability $\boldsymbol{p}_X$, which is assumed to follow the above Dirichlet distribution, thus deriving its optimization goal as

$$\begin{aligned} \mathcal{L}_{edl} &= \frac{1}{|\mathcal{D}|} \sum_{(\boldsymbol{z}, \boldsymbol{y}) \in \mathcal{D}} \mathbb{E}_{\boldsymbol{p}_X \sim \mathrm{Dir}(\boldsymbol{p}_X, \boldsymbol{\alpha}_X)} \left[ \|\boldsymbol{y} - \boldsymbol{p}_X\|_2^2 \right] \\ &= \frac{1}{|\mathcal{D}|} \sum_{(\boldsymbol{z}, \boldsymbol{y}) \in \mathcal{D}} \sum_{x \in \mathbb{X}} \left( \boldsymbol{y}[x] - \mathbb{E}_{\boldsymbol{p}_X \sim \mathrm{Dir}(\boldsymbol{p}_X, \boldsymbol{\alpha}_X)}[\boldsymbol{p}_X(x)] \right)^2 + \mathrm{Var}_{\boldsymbol{p}_X \sim \mathrm{Dir}(\boldsymbol{p}_X, \boldsymbol{\alpha}_X)}[\boldsymbol{p}_X(x)], \end{aligned} \tag{2}$$

where the training set $\mathcal{D}$ consists of sample features and their one-hot labels denoted $(\boldsymbol{z}, \boldsymbol{y})$[1], and $\boldsymbol{y}[x]$ refers to the $x$-th element of $\boldsymbol{y}$. A rigorous mathematical exposition of subjective logic and a more detailed introduction of EDL will be provided in section 2 and section 3.1.

Despite the remarkable success of EDL, we argue that in the existing EDL-based methodology, there exists nonessential settings in both model construction and model optimization. These settings have been widely accepted by deep learning researchers, however, they are not intrinsically mandated in the mathematical framework of subjective logic. Specifically, **(1)** in model construction, the commonly ignored prior weight parameter in Eqn. 1 actually governs the balance between capitalizing on the proportion of evidence and its magnitude when deriving predictive scores. However, EDL prescribes this parameter's value to be equivalent to the number of classes, potentially resulting in highly counter-intuitive outcomes. Therefore, we advocate for setting the prior weight parameter as a free hyper-parameter in the neural network to adapt to complex application cases. **(2)** in model optimization, the EDL loss function given by Eqn. 2 includes a variance-minimized regularization term, which encourages the Dirichlet PDF modeling the distribution of probabilities to approach a Dirac delta function which is infinitely high and infinitesimally thin, or in other words, requires an infinite amount of evidence of the target class, thus further intensifying the over-confidence issue. Contrarily, we advocate for directly optimizing the expectation of the Dirichlet distribution towards the given one-hot label, thus deprecating this regularization to obtain more reliable predictive scores. Note that both the above relaxations strictly adhere to the subjective logic theory. Theoretical analysis in section 3 and experiment results in section 5 both demonstrate that relaxing the above nonessential settings contributes to alleviating the over-confidence issue and bringing more accurate uncertainty estimation. Our contributions include:

- An analysis of the significance of the commonly ignored parameter termed prior weight on balancing the trade-off relationship between leveraging the proportion and magnitude of evidence to compute predictive scores in the subjective logic framework. Relaxing the rigid setting of fixing the parameter to the number of classes has been shown to enhance the quality of uncertainty estimation.

- An analysis of the advantages of directly optimizing the expected value of the constructed Dirichlet distribution, instead of minimizing the integration of MSE loss over the class probability $\boldsymbol{p}_X$ which follows the above Dirichlet distribution. Relaxing the EDL optimization objective by deprecating the variance-minimized regularization term has been shown to mitigate the issue of over-confidence.

- Extensive experiments on multiple benchmarks for uncertainty estimation tasks, including confidence estimation and out-of-distribution detection, which comprehensively demonstrate the effectiveness of our proposed R-EDL with remarkable performances under the classical, few-shot, noisy, and video-modality settings.

Derivations, proofs, additional experiment results and details are given in Appendix.

---

[1] In deep learning, the sample feature is usually denoted by the symbol $\boldsymbol{x}$. However, to preclude ambiguity with the symbol $x$ denoting the value of the random variable $X$, we employ $\boldsymbol{z}$ instead of $\boldsymbol{x}$ to denote the sample feature. The random variable $X$, the label $\boldsymbol{y}$, and the feature $\boldsymbol{z}$ pertain to the same input sample.

## 2 SUBJECTIVE LOGIC THEORY

Just as the names of *binary* logic and *probabilistic* logic imply, an argument in binary logic must be either true or false, and one in probabilistic logic can take its probability in the range $[0, 1]$ to express the meaning of partially true. Furthermore, subjective logic (Jøsang, 2001; 2016) extends probabilistic logic by explicitly including uncertainty about probabilities in the formalism. Specifically, an argument in subjective logic, also called a *subjective opinion*, is formalized as follows:

**Definition 1 (Subjective opinion).** Let $X$ be a categorical random variable on the domain $\mathbb{X}$. A subjective opinion over the random variable $X$ is defined as the ordered triplet $\boldsymbol{\omega}_X = (\boldsymbol{b}_X, u_X, \boldsymbol{a}_X)$, where $\boldsymbol{b}_X$ is a *belief mass* distribution over $X$, $u_X$ is a *uncertainty mass*, $\boldsymbol{a}_X$ is a *base rate*, aka prior probability distribution over $X$, and the additivity requirements $\sum_{x \in \mathbb{X}} \boldsymbol{b}_X(x) + u_X = 1$ and $\sum_{x \in \mathbb{X}} \boldsymbol{a}_X(x) = 1$ are satisfied.

Belief mass assigned to a singleton value $x \in \mathbb{X}$ expresses support for $x$ being TRUE, and uncertainty mass can be interpreted as belief mass assigned to the entire domain. Therefore, subjective logic also provides a well-defined *projected probability*, which follows the additivity requirement of traditional probability theory, by reassigning the uncertainty mass into each singleton of domain $\mathbb{X}$ according to the base rate $\boldsymbol{a}_X$ as follows:

**Definition 2 (Projected probability of a subjective opinion).** Let $\boldsymbol{\omega}_X = (\boldsymbol{b}_X, u_X, \boldsymbol{a}_X)$ be a subjective opinion. The projected probability $\boldsymbol{P}_X$ of the opinion $\boldsymbol{\omega}_X$ is defined by $\boldsymbol{P}_X(x) = \boldsymbol{b}_X(x) + \boldsymbol{a}_X(x)u_X, \forall x \in \mathbb{X}$. Note that the additivity requirement $\sum_{x \in \mathbb{X}} \boldsymbol{P}_X(x) = 1$ is satisfied.

Furthermore, the subjective logic theory points out that, if the base rate $\boldsymbol{a}_X$ and a parameter termed prior weight, denoted as $W$, is given, there exists a bijection between a multinomial opinion and a Dirichlet probabilistic density function (PDF). This relationship emerges from interpreting second-order uncertainty by probability density, and plays an important role in the formalism of subjective logic since it provides a calculus reasoning with PDFs. The proof is provided in Appendix A.1.

**Theorem 1 (Bijection between subjective opinions and Dirichlet PDFs).** Let $X$ be a random variable defined in domain $\mathbb{X}$, and $\boldsymbol{\omega}_X = (\boldsymbol{b}_X, u_X, \boldsymbol{a}_X)$ be a subjective opinion. $\boldsymbol{p}_X$ is a probability distribution over $\mathbb{X}$, and a Dirichlet PDF with the concentration parameter $\boldsymbol{\alpha}_X$ is denoted by $\mathrm{Dir}(\boldsymbol{p}_X, \boldsymbol{\alpha}_X)$, where $\boldsymbol{\alpha}_X(x) \geq 0$, and $\boldsymbol{p}_X(x) \neq 0$ if $\boldsymbol{\alpha}_X(x) < 1$. Then, given the base rate $\boldsymbol{a}_X$, there exists a bijection $F$ between the opinion $\boldsymbol{\omega}_X$ and the Dirichlet PDF $\mathrm{Dir}(\boldsymbol{p}_X, \boldsymbol{\alpha}_X)$:

$$F : \boldsymbol{\omega}_X = (\boldsymbol{b}_X, u_X, \boldsymbol{a}_X) \mapsto \mathrm{Dir}(\boldsymbol{p}_X, \boldsymbol{\alpha}_X) = \frac{\Gamma\left(\sum_{x \in \mathbb{X}} \boldsymbol{\alpha}_X(x)\right)}{\prod_{x \in \mathbb{X}} \Gamma(\boldsymbol{\alpha}_X(x))} \prod_{x \in \mathbb{X}} \boldsymbol{p}_X(x)^{\boldsymbol{\alpha}_X(x) - 1}, \quad (3)$$

where $\Gamma$ denotes the Gamma function, $\boldsymbol{\alpha}_X$ satisfies the following identity that

$$\boldsymbol{\alpha}_X(x) = \frac{\boldsymbol{b}_X(x)W}{u_X} + \boldsymbol{a}_X(x)W, \quad \forall x \in \mathbb{X}, \quad (4)$$

and $W \in \mathbb{R}_+$ is a scalar called a prior weight, whose setting will be further discussed in section 3.2.

## 3 R-EDL: ALLEVIATING OVER-CONFIDENCE BY RELAXING NONESSENTIAL SETTINGS OF EDL

Despite the significant success achieved by EDL and its related works, we argue that the existing EDL-based methodology (section 3.1) keeps rigid settings on the construction of the Dirichlet distributions specified in Theorem 1 and the design of optimization objectives, which, while widely accepted, are not intrinsically mandated within the subjective logic framework (section 2). Theoretical analysis in this section and comprehensive experiments in section 5 both demonstrate that those nonessential settings hinder this line of methods from quantifying more accurate uncertainty. Specifically, in this section, we rigorously analyze and relax two nonessential settings in EDL, including: (1) in model construction, the prior weight parameter is prescribed to be equivalent to the number of classes (section 3.2); (2) in model optimization, the traditional optimization objective includes a variance-minimized regularization term, which potentially intensifies over-confidence (section 3.3). Note that our relaxations to the above EDL settings strictly adhere to subjective logic.

### 3.1 PRELIMINARY: EVIDENTIAL DEEP LEARNING

Based on the subjective logic theory, Sensoy et al. (2018) proposes a single-forward-pass uncertainty estimation method named Evidential Deep Learning (EDL), which lets deep neural networks play

the role of analysts to give belief mass and uncertainty mass of samples. For example, in the case of $C$-class classification, the belief mass $\boldsymbol{b}_X$ and uncertainty mass $u_X$ of the input sample, whose category index is a random variable $X$ taking values $x$ from the domain $\mathbb{X} = [1, ..., C]$, are given by

$$\boldsymbol{b}_X(x) = \frac{\boldsymbol{e}_X(x)}{\sum_{x' \in \mathbb{X}} \boldsymbol{e}_X(x') + C}, \quad u_X = \frac{C}{\sum_{x \in \mathbb{X}} \boldsymbol{e}_X(x) + C}, \quad \forall x \in \mathbb{X}. \tag{5}$$

Specifically, $\boldsymbol{e}_X(x)$, which denotes the *evidence* of the random variable $X$ taking the value $x$, is the $x$-th element of the evidence vector $\boldsymbol{e}_X = f(g(\boldsymbol{z})) \in \mathbb{R}_+^C$, where $\boldsymbol{z}$ is the feature of the input sample, $g$ is a deep neural network, $f$ is a non-negative activation function, e.g., softplus, sometimes also called the evidence function, and the scalar $C$ in this equation serves as the prior weight.

According to Theorem 1, there exists a bijection between the Dirichlet PDF denoted $\text{Dir}_X(\boldsymbol{p}_X, \boldsymbol{\alpha}_X)$ and the opinion $\boldsymbol{\omega}_X = (\boldsymbol{b}_X, u_X, \boldsymbol{a}_X)$ if the requirement in Eqn. 4 is satisfied. Substituting Eqn. 5 into Eqn. 4 and setting the prior weight $W$ in Eqn. 4 as $C$, we obtain the relationship between the parameter vector of the Dirichlet PDF and the collected evidence in EDL, as expressed by Eqn. 1. Moreover, since EDL sets the base rate $\boldsymbol{a}_X(x)$ as a uniform distribution, the relationship given by Eqn. 1 can be further simplified into $\boldsymbol{\alpha}_X(x) = \boldsymbol{e}_X(x) + 1, \forall x \in \mathbb{X}$.

To perform model optimization, EDL integrates the conventional MSE loss function over the class probability $\boldsymbol{p}_X$ which is assumed to follow the Dirichlet PDF specified in the bijection, thus derives the optimization objective given by Eqn. 2. The detailed derivation is provided in Appendix A.2. In inference, EDL utilizes the projected probability $\boldsymbol{P}_X$ (refer to Definition 2) as the predictive scores, and uses Eqn. 5 to calculate the uncertainty mass $u_X$ as the uncertainty of classification,

$$\boldsymbol{P}_X(x) = \frac{\boldsymbol{e}_X(x) + 1}{\sum_{x' \in \mathbb{X}} \boldsymbol{e}_X(x') + C} = \frac{\boldsymbol{\alpha}_X(x)}{S_X}, \quad u_X = \frac{C}{\sum_{x \in \mathbb{X}} \boldsymbol{e}_X(x) + C} = \frac{C}{S_X}, \quad \forall x \in \mathbb{X}, \tag{6}$$

where $S_X$ is the sum of $\boldsymbol{\alpha}_X(x)$ over $x \in \mathbb{X}$.

## 3.2 Relaxing Rigid Setting of Prior Weight in Model Construction

In this subsection, we elucidate how $W$ orchestrates the equilibrium between leveraging the proportion and magnitude of evidence to compute predictive scores. Conclusively, we argue against the rigidity of fixing $W$ to the class number and propose viewing it as an adjustable hyper-parameter.

The nomenclature of prior weight comes from the expression of Eqn. 1. Here, the scalar coefficient $C$, functioning as the prior weight $W$, denotes the weight of the base rate $\boldsymbol{a}_X$, which is alternatively termed the prior distribution. In Theorem 1, it should be noted that the existence of the bijection is contingent upon certain prerequisites; specifically, the base rate $\boldsymbol{a}_X$ and the prior weight $W$ must be provided. Typically, in the absence of prior information, we default to setting the base rate as a uniform distribution over the domain $\mathbb{X}$, i.e., $\boldsymbol{a}_X(x) = 1/C, \forall x \in \mathbb{X}$, and $|\mathbb{X}| = C$. However, the setting of the prior weight $W$ is worth further discussion.

We argue that fixing the prior weight to the cardinality of the domain, which is widely adopted by EDL researchers, is not intrinsically mandated by subjective logic and may result in counter-intuitive results. For example, a 100-classes classification task forces $W = 100$. Even though the neural net gives an extreme evidence distribution $\boldsymbol{e} = [100, 0, 0, ...., 0] \in \mathbb{R}_+^{100}$, EDL will reach the prediction that the probability of the sample belonging to Class 1 is $P = (100 + 1)/(100 + 100) \approx 0.5$ by Eqn. 6, which is highly counter-intuitive. The underlying reason for the above phenomenon is that the value of $W$ dictates the degree to which the projected probability is influenced by the magnitude of the evidence or contrarily the proportion of the evidence. To elucidate this point more clearly, we first revisit Eqn. 5 and Eqn. 6 without fixing the prior weight $W$ to $C$. In this way, we can obtain a generalized form of the projected probability $\boldsymbol{P}_X$ as

$$\boldsymbol{P}_X(x) = \boldsymbol{b}_X(x) + \boldsymbol{a}_X(x) u_X = \frac{\boldsymbol{e}_X(x) + \frac{W}{C}}{\sum_{x' \in \mathbb{X}} \boldsymbol{e}_X(x') + W}, \quad \forall x \in \mathbb{X}. \tag{7}$$

When the prior weight $W$ is set to zero, the projected probability $\boldsymbol{P}_X$ in Eqn. 7 degenerates to a conventional probability form, which solely relies on the proportion of evidence among classes and is unaffected by their magnitude, as scaling the evidence by a constant coefficient has no impact on $\boldsymbol{P}_X$. However, when $W$ is not zero, we have

$$\boldsymbol{P}_X(x) \leq \frac{\boldsymbol{e}_X(x) + \frac{W}{C}}{\boldsymbol{e}_X(x) + W} = 1 - (1 - \frac{1}{C}) \cdot \frac{1}{\boldsymbol{e}_X(x)/W + 1}, \quad \forall x \in \mathbb{X}, \tag{8}$$

where the equlity holds if $\sum_{x' \in \mathbb{X}, x' \neq x} \boldsymbol{e}_X(x') = 0$. Eqn. 8 indicates that, in scenarios of extreme evidence distributions, i.e., when the evidence for all classes except class $x$ is zero, the upper bound of $\boldsymbol{P}_X(x)$ is governed by the ratio of the evidence for class $x$ to the prior weight $W$. In other words, the upper bound of $\boldsymbol{P}_X(x)$ purely relies on the magnitude of $\boldsymbol{e}_X(x)$ when the prior weight $W$ is given, and a lower magnitude results in a larger gap between the upper bound of $\boldsymbol{P}_X(x)$ and 1.

From the two cases presented above, it becomes evident that the value of $W$ determines the extent to which the projected probability $\boldsymbol{P}_X(x)$ is influenced by the magnitude and proportion of evidence respectively. Specifically, a small $W$ implies that $\boldsymbol{P}_X(x)$ is predominantly influenced by the proportion of evidence distribution, whereas a large $W$ leads $\boldsymbol{P}_X(x)$ to mainly considering the magnitude of the evidence while overlooking the evidence proportion.

Intuitively speaking, for any specific case, there should exist an optimal value for $W$ which can balance the inherent trade-off between leveraging the proportion of evidence and its magnitude to obtain predictive scores minimizing the model over-confidence on misclassified and out-of-distribution samples. However, it is unlikely that such an optimal value is universally applicable to all scenarios, given the myriad of complex factors influencing the network's output. Hence, we advocate for relinquishing the rigidity of assigning the number of classes to $W$, but instead, treating $W$ as an adjustable hyper-parameter within the neural network. Therefore, we revisit Eqn. 4 to derive a generalized form of the concentration parameter $\boldsymbol{\alpha}_X$ of the constructed Dirichlet PDF as

$$\boldsymbol{\alpha}_X(x) = \left( \frac{\boldsymbol{e}_X(x)}{W} + \frac{1}{|\mathbb{X}|} \right) W = \boldsymbol{e}_X(x) + \lambda, \quad \forall x \in \mathbb{X}, \tag{9}$$

where $\lambda = W/C \in \mathbb{R}_+$ is a hyper-parameter. Note that both the projected probability and the uncertainty mass retain the same form as in Eqn. 6, i.e., $\boldsymbol{P}_X(x) = \boldsymbol{\alpha}_X(x)/S_X$ and $u_X = C/S_X$, when represented by $\boldsymbol{\alpha}_X(x)$ and $S_X$.

### 3.3 Deprecating Variance-minimized Regularization in Model Optimization

In the preceding subsection, we underscore the imperative of treating the prior weight $W$ as an adjustable hyper-parameter, which enables the projected probability $\boldsymbol{P}_X$ to effectively balance the trade-off between leveraging the proportion and the magnitude of collected evidence. Consequently, in this subsection, we elucidate the reasoning underlying our optimization objective, which focuses on directly optimizing the projected probability $\boldsymbol{P}_X$. Upon comparison with the traditional loss function employed in EDL, it becomes evident that our method deprecates a commonly used variance-minimizing regularization term. We undertake a meticulous examination of the motivations for relaxing the EDL optimization objective by excluding this term.

With the generalized setting of $\boldsymbol{\alpha}_X$ in Eqn. 9, the projected probability $\boldsymbol{P}_X$ has the following variant:

$$\boldsymbol{P}_X(x) = \frac{\boldsymbol{\alpha}_X(x)}{S_X} = \frac{\boldsymbol{e}_X(x) + \lambda}{\sum_{x' \in \mathbb{X}} \boldsymbol{e}_X(x') + C\lambda}, \quad \forall x \in \mathbb{X}. \tag{10}$$

Consequently, by substituting the class probability in traditional MSE loss with the projected probability $\boldsymbol{P}_X$ in Eqn. 10, we seamlessly derive an appropriate optimization objective denoted $\mathcal{L}_{redl}$ within our relaxed-EDL framework in the following form:

$$\mathcal{L}_{redl} = \frac{1}{|\mathcal{D}|} \sum_{(\boldsymbol{z}, \boldsymbol{y}) \in \mathcal{D}} \sum_{x \in \mathbb{X}} \left( \boldsymbol{y}[x] - \boldsymbol{P}_X(x) \right)^2. \tag{11}$$

Regarding the reason for adopting the above optimization objective, we contend that the projected probability $\boldsymbol{P}_X$ has the unique property of alleviating the overconfidence typically arising from optimization toward the hard one-hot labels $\boldsymbol{y}$. As previously noted, the projected probability $\boldsymbol{P}_X$ harnesses both the magnitude and proportion of collected evidence to more accurately represent the actual likelihood of a given output. From an optimization perspective, compared to the proportion of evidence among classes, i.e., $\boldsymbol{e}_X(x)/\sum_x \boldsymbol{e}_X(x)$, or the belief mass $\boldsymbol{b}_X$, the projected probability $\boldsymbol{P}_X$ has more tolerance towards the existence of the uncertainty mass $u_X$, since $u_X$ also contributes to the projected probability $\boldsymbol{P}_X$ according to the base rate $\boldsymbol{a}_X$. In other words, the item $\boldsymbol{a}_X u_X$ alleviates the urgency of the projected probability $\boldsymbol{P}_X$ tending to the one-hot label $\boldsymbol{y}$ when the model has not collected enough evidence, since the uncertainty mass $u_X$ is inversely proportional to the total amount of evidence, thus mitigating the over-confidence issue to some extent.

Meanwhile, the optimization goal in Eqn. 11 can also be interpreted as encouraging the expectation of the Dirichlet distribution to converge to the provided label, since the bijection introduced in Theorem 1 has been established on the following identity:

$$\boldsymbol{P}_X(x) = \mathbb{E}_{\boldsymbol{p}_X \sim \mathrm{Dir}(\boldsymbol{p}, \boldsymbol{\alpha})}[\boldsymbol{p}_X(x)], \tag{12}$$

which can be easily derived from Eqn. 10 and the property of Dirichlet distributions. Therefore, by substituting Eqn. 12 into Eqn. 11 and then comparing it with Eqn. 2, we can find that the essential difference between the two optimization goals is that, EDL optimizes the expectation of the traditional MSE loss over the constructed Dirichlet PDF, while our proposed R-EDL directly optimizes the expectation of the constructed Dirichlet PDF with MSE loss. As a result, a regularization term, denoted $\mathcal{L}_{var}$, which attempts to minimize the variance of the Dirichlet distribution given by the following equation is deprecated:

$$\mathcal{L}_{var} = \frac{1}{|\mathcal{D}|} \sum_{(\boldsymbol{z},\boldsymbol{y}) \in \mathcal{D}} \sum_{x \in \mathbb{X}} \mathrm{Var}_{\boldsymbol{p}_X \sim \mathrm{Dir}(\boldsymbol{p}_X, \boldsymbol{\alpha}_X)}[\boldsymbol{p}_X(x)] = \frac{1}{|\mathcal{D}|} \sum_{(\boldsymbol{z},\boldsymbol{y}) \in \mathcal{D}} \frac{S_X^2 - \sum_{x \in \mathbb{X}} \boldsymbol{\alpha}_X^2(x)}{S_X^2(S_X + 1)}. \tag{13}$$

Let us delve deeper into this variance-minimized regularization term. When the variance of a Dirichlet distribution is close to zero, the Dirichlet probability density function is in the form of a Dirac delta function which is infinitely high and infinitesimally thin. Consequently, in the entire training phase, the regularization term $\mathcal{L}_{var}$ keeps requiring an infinite amount of evidence of the target class, which further intensifies the serious over-confidence issue we seek to mitigate. From another perspective, the Dirichlet distribution which models the distribution of first-order probabilities would gradually degenerate to a traditional point estimation of first-order probabilities when its variance approaches zero, thus losing the advantage of subjective logic in modeling second-order uncertainty. Therefore, we posit that omitting this regularization term contributes to alleviating the over-confidence issue which commonly results in suboptimal uncertainty estimation, while preserving the merits of subjective logic. Our ablation study further corroborates this assertion. Moreover, following previous works (Sensoy et al., 2018; Deng et al., 2023), we adopt an additional KL-divergence based regularization for optimization, and its detailed introduction can be found in Appendix A.2.

## 4 RELATED WORK

**Extensions and applications of EDL.** A detailed introduction of EDL can be found in section 3.1, and here we briefly introduce the follow-up works of EDL. After Sensoy et al. (2018) proposes EDL, Deep Evidential Regression (DER) (Amini et al., 2020; Soleimany et al., 2021) extend this paradigm by incorporating evidential priors into the conventional Gaussian likelihood function, thereby enhancing the modeling of uncertainty within regression networks. Kandemir et al. (2022) combines EDL, neural processes, and neural Turing machines to propose the Evidential Tuning Process, which shows stronger performances than EDL but requires a rather complex memory mechanism. Meinert et al. (2023) offers further insights into the empirical effectiveness of DER, even in the presence of over-parameterized representations of uncertainty. Recently, $\mathcal{I}$-EDL proposed by Deng et al. (2023) largely outperforms EDL by incorporating Fisher information matrix to measure the informativeness of evidence carried by samples. For application, DEAR (Bao et al., 2021) achieves impressive performances on open-set action recognition by proposing a novel model calibration method to regularize the EDL training. Moreover, EDL has achieved great success in other applications of computer vision (Qin et al., 2022; Oh & Shin, 2022; Sun et al., 2022; Park et al., 2022; Sapkota & Yu, 2022; Chen et al., 2023a;b; Gao et al., 2023b). Compared with previous efforts, our method is the first to consider relaxing the nonessential settings of the traditional EDL while strictly adhering to the subjective logic theory.

**Other single-model uncertainty methods based on DNNs.** In addition to EDL-related works, various single-model methods exist for estimating predictive uncertainties. Efficient ensemble methods (Wen et al., 2020; Dusenberry et al., 2020), which cast a set of models under a single one, show state-of-the-art performances on large-scale datesets. While these methods are parameter-efficient, they necessitate multiple forward passes during inference. Bayesian Neural Networks (BNNs)(Ritter et al., 2018; Izmailov et al., 2021) model network parameters as random variables and quantify uncertainty through posterior estimation while suffering from a significant computational cost. A widely-recognized method is Monte Carlo Dropout (Gal & Ghahramani, 2016), which interprets the dropout layer as a random variable following a Bernoulli distribution, and training a neural network

with such dropout layers can be considered an approximation to variational inference. Two other notable single-forward-pass methods, DUQ (Van Amersfoort et al., 2020) and SNGP (Liu et al., 2020), introduce distance-aware output layers using radial basis functions or Gaussian processes. Although nearly competitive with deep ensembles in OOD benchmarks, these methods entail extensive modifications to the training procedure and lack easy integration with existing classifiers. Another group of efficient uncertainty methods are Dirichlet-based uncertainty (DBU) methods, to which EDL also belongs. Prominent DBU methods encompass KL-PN (Malinin & Gales, 2018), RKL-PN (Malinin & Gales, 2019), and Posterior Network (Charpentier et al., 2020), which vary in both the parameterization and the training strategy of the Dirichlet distribution. Compared to these preceding methods, our approach combines the benefits of exhibiting favorable performances, being single-forward-pass, parameter-efficient, and easily integrable.

## 5 EXPERIMENTS

### 5.1 EXPERIMENTAL SETUP

**Baselines.** Following Deng et al. (2023), we focus on comparing with other Dirichlet-based uncertainty methods, including the traditional **EDL** (Sensoy et al., 2018), $\mathcal{I}$-**EDL** (Deng et al., 2023), **KL-PN** (Malinin & Gales, 2018), **RKL-PN** (Malinin & Gales, 2019), and **PostN** (Charpentier et al., 2020). Additionally, we present the results of the representative single-forward-pass method **DUQ** (Van Amersfoort et al., 2020) and the popular Bayesian uncertainty method **MC Dropout** (Gal & Ghahramani, 2016) for reference. For experiments concerning video-modality data, following Bao et al. (2021), we compare our methods with: **OpenMax** (Bendale & Boult, 2016), **MC Dropout**, **BNN SVI** (Krishnan et al., 2018), **RPL** (Chen et al., 2020), and **DEAR** (Bao et al., 2021).

**Datasets, Implementation details, Hyper-parameter settings.** Refer to Appendix C.

### 5.2 CLASSICAL SETTING

A classifier with reliable uncertainty estimation abilities should exhibit following characteristics: (1) Assign higher uncertainties to out-of-distribution (OOD) than in-distribution (ID) samples; (2) Assign higher uncertainties to misclassified than to correctly classified samples; (3) maintain comparable classification accuracy. Therefore, We first evaluate our method by OOD detection and confidence estimation in image classification, measured by the area under the precision-recall curve (**AUPR**) with labels 1 for ID / correctly classified data, and labels 0 for OOD / misclassified data. For the Dirichlet-base uncertainty methods, we use the **max probability** (**MP**) and the sum of Dirichlet concentration parameters, aka the scaled reciprocal of **uncertainty mass** (**UM**) of subjective opinions, as the confidence scores. For MC Dropout and DUQ, we only report their MP performances since they do not involve Dirichlet PDFs. As Table 1 shows, our R-EDL method shows consistently favorable performances on most metrics. In particular, comparing with the traditional EDL method and the SOTA method $\mathcal{I}$-EDL, our R-EDL obtains absolute gains of 6.13% and 1.74% when evaluated by MP on the OOD detection setting of CIFAR-10 against SVHN. Besides, our method also achieves superior performances on confidence estimation while maintaining a satisfactory classification accuracy. All results are averaged from 5 runs, and the relatively small standard deviations indicate that R-EDL exhibits stable performances.

### 5.3 FEW-SHOT SETTING

Next, we conduct more challenging few-shot experiments on mini-ImageNet to further demonstrate the effectiveness of our method. As shown in Table 2, we report the averaged top-1 accuracy of classification and the AUPR scores of confidence estimation and OOD detection over 10,000 few-shot episodes. We employ the $N$-way $K$-shot setting, with $N \in \{5, 10\}$ and $K \in \{1, 5, 20\}$. Each episode comprises $N$ random classes and $K$ random samples per class for training, $\min(15, K)$ query samples per class for classification and confidence estimation, and an equivalent number of query samples from the CUB dataset for OOD detection. As depicted in Table 2, our R-EDL method achieves satisfactory performances on most $N$-way $K$-shot settings. Specifically, comparing with the EDL and $\mathcal{I}$-EDL methods, our R-EDL obtains absolute gains of 9.19% and 1.61% when evaluated by MP on OOD detection of the 5-way 5-shot task.

### 5.4 NOISY SETTING

Thereafter, we employ noisy data to assess both the robustness of classification and the OOD detection capability of our method in the presence of noise. Following Deng et al. (2023), we generate

Table 1: Accuracy of classification and AUPR scores of confidence estimation and OOD detection, averaged over 5 runs. On MNIST we adopt ConvNet consisting of 3 conventional layers and 3 dense layers as the backbone, while on CIFAR10 we adopt VGG16. A→B denotes taking A/B as ID/OOD data. MP refers to max probability, and UM refers to uncertainty mass.

| Method | MNIST | | | MNIST→KMNIST | | MNIST→FMNIST | |
| | Classification | Confidence Estimation | | OOD Detection | | OOD Detection | |
| | Acc | MP | UM | MP | UM | MP | UM |
|---|---|---|---|---|---|---|---|
| MC Dropout | 99.26±0.0 | 99.98±0.0 | - | 94.00±0.1 | - | 96.56±0.3 | - |
| DUQ | 98.65±0.1 | 99.97±0.0 | - | 98.52±0.1 | - | 97.92±0.6 | - |
| KL-PN | 99.01±0.0 | 99.92±0.0 | 99.95±0.0 | 92.97±1.2 | 93.39±1.0 | 98.44±0.1 | 98.16±0.0 |
| RKL-PN | 99.21±0.0 | 99.67±0.0 | 99.57±0.0 | 60.76±2.9 | 53.76±3.4 | 78.45±3.1 | 72.18±3.6 |
| PostN | **99.34±0.0** | 99.98±0.0 | 99.97±0.0 | 95.75±0.2 | 94.59±0.3 | 97.78±0.2 | 97.24±0.3 |
| EDL | 98.22±0.31 | 99.99±0.00 | 99.98±0.00 | 97.02±0.76 | 96.31±2.03 | 98.10±0.44 | 98.08±0.42 |
| $\mathcal{I}$-EDL | 99.21±0.08 | 99.98±0.00 | 99.98±0.00 | 98.34±0.24 | 98.33±0.24 | 98.89±0.28 | 98.86±0.29 |
| R-EDL(Ours) | 99.33±0.03 | **99.99±0.00** | **99.99±0.00** | **98.69±0.19** | **98.69±0.20** | **99.29±0.11** | **99.29±0.12** |

| Method | CIFAR10 | | | CIFAR10→SVHN | | CIFAR10→CIFAR100 | |
| | Classification | Confidence Estimation | | OOD Detection | | OOD Detection | |
| | Acc | MP | UM | MP | UM | MP | UM |
|---|---|---|---|---|---|---|---|
| MC Dropout | 82.84±0.1 | 97.15±0.0 | - | 51.39±0.1 | - | 45.57±1.0 | - |
| DUQ | 89.33±0.2 | 97.89±0.3 | - | 80.23±3.4 | - | 84.75±1.1 | - |
| KL-PN | 27.46±1.7 | 50.61±4.0 | 52.49±4.2 | 43.96±1.9 | 43.23±2.3 | 61.41±2.8 | 61.53±3.4 |
| RKL-PN | 64.76±0.3 | 86.11±0.4 | 85.59±0.3 | 53.61±1.1 | 49.37±0.8 | 55.42±2.6 | 54.74±2.8 |
| PostN | 84.85±0.0 | 97.76±0.0 | 97.25±0.0 | 80.21±0.2 | 77.71±0.3 | 81.96±0.8 | 82.06±0.8 |
| EDL | 83.55±0.64 | 97.86±0.17 | 97.83±0.17 | 78.87±3.50 | 79.12±3.69 | 84.30±0.67 | 84.18±0.74 |
| $\mathcal{I}$-EDL | 89.20±0.32 | 98.72±0.12 | 98.63±0.11 | 83.26±2.44 | 82.96±2.17 | 85.35±0.69 | 84.84±0.64 |
| R-EDL(Ours) | **90.09±0.30** | **98.98±0.05** | **98.98±0.05** | **85.00±1.22** | **85.00±1.22** | **87.72±0.31** | **87.73±0.31** |

Table 2: Results of the few-shot setting for WideResNet-28-10 on mini-ImageNet, with CUB as OOD data, averaged over 10000 episodes.

| Method | 5-Way 1-Shot | | | | | 10-Way 1-Shot | | | | |
| | Classification | Confidence Estimation | | OOD Detection | | Classification | Confidence Estimation | | OOD Detection | |
| | Acc | MP | UM | MP | UM | Acc | MP | UM | MP | UM |
|---|---|---|---|---|---|---|---|---|---|---|
| EDL | 61.00±0.11 | 78.95±0.12 | 75.34±0.12 | 66.78±0.12 | 65.41±0.13 | 44.55±0.08 | 63.37±0.11 | 61.68±0.10 | 59.19±0.09 | 67.81±0.12 |
| $\mathcal{I}$-EDL | 63.82±0.10 | **82.83±0.11** | 80.33±0.11 | 71.79±0.12 | 74.76±0.13 | 49.37±0.07 | 68.40±0.09 | **67.54±0.09** | 71.60±0.10 | 71.95±0.10 |
| R-EDL(Ours) | **63.93±0.11** | 82.75±0.11 | **80.80±0.11** | **72.91±0.12** | **74.84±0.13** | **50.02±0.07** | **68.68±0.09** | 67.12±0.09 | **72.83±0.10** | **73.08±0.10** |

| Method | 5-Way 5-Shot | | | | | 10-Way 5-Shot | | | | |
| | Classification | Confidence Estimation | | OOD Detection | | Classification | Confidence Estimation | | OOD Detection | |
| | Acc | MP | UM | MP | UM | Acc | MP | UM | MP | UM |
|---|---|---|---|---|---|---|---|---|---|---|
| EDL | 80.38±0.08 | 94.30±0.04 | 92.09±0.05 | 74.46±0.10 | 76.53±0.14 | 62.50±0.08 | 87.55±0.05 | 84.35±0.06 | 71.06±0.10 | 76.28±0.10 |
| $\mathcal{I}$-EDL | **82.00±0.07** | **94.42±0.04** | 93.61±0.05 | 82.04±0.10 | 82.48±0.10 | 67.89±0.06 | **89.14±0.04** | 85.52±0.05 | 80.63±0.11 | 82.29±0.10 |
| R-EDL(Ours) | 81.85±0.07 | 94.29±0.04 | **93.65±0.05** | **83.65±0.10** | **84.22±0.10** | **70.51±0.05** | 88.84±0.04 | **86.26±0.05** | **82.39±0.09** | **83.37±0.09** |

| Method | 5-Way 20-Shot | | | | | 10-Way 20-Shot | | | | |
| | Classification | Confidence Estimation | | OOD Detection | | Classification | Confidence Estimation | | OOD Detection | |
| | Acc | MP | UM | MP | UM | Acc | MP | UM | MP | UM |
|---|---|---|---|---|---|---|---|---|---|---|
| EDL | 85.54±0.06 | 97.77±0.02 | 97.05±0.02 | 80.01±0.10 | 79.78±0.12 | 69.30±0.09 | 94.57±0.03 | 93.15±0.03 | 74.50±0.08 | 76.89±0.09 |
| $\mathcal{I}$-EDL | 88.12±0.05 | **97.93±0.02** | 96.98±0.02 | 84.29±0.09 | 85.40±0.09 | 78.59±0.04 | **95.15±0.02** | 93.32±0.03 | 81.34±0.07 | 82.52±0.07 |
| R-EDL(Ours) | **88.74±0.05** | 97.83±0.02 | **97.10±0.02** | **84.85±0.09** | **85.57±0.09** | **79.79±0.04** | 94.81±0.02 | **93.47±0.02** | **82.22±0.08** | **82.72±0.07** |

the noisy data by introducing zero-mean isotropic Gaussian noise to the test split of the ID dataset. Fig. 1(a) clearly illustrates the superior performance of R-EDL in terms of the average of these two key metrics. More results and analysis are provided in Appendix D.3.

## 5.5 ABLATION STUDY AND PARAMETER ANALYSIS

We assess the performance impact of relaxing two aforementioned nonessential settings in EDL, as summarized in Table 3. In particular, we explore the effects of retaining the original value of $\lambda = 1$, and of reintroducing the deprecated variance-minimized regularization term $\mathcal{L}_{var}$. Note that if both original settings are restored, R-EDL reverts to traditional EDL. As evidenced in rows 3 and 4 of Table 3, reverting to each of these original settings results in a noticeable performance decline, or conversely, relaxing these settings leads to performance gains, particularly in OOD detection. For instance, measured by the AUPR score for OOD detection in the setting of CIFAR-10 vs SVHN,

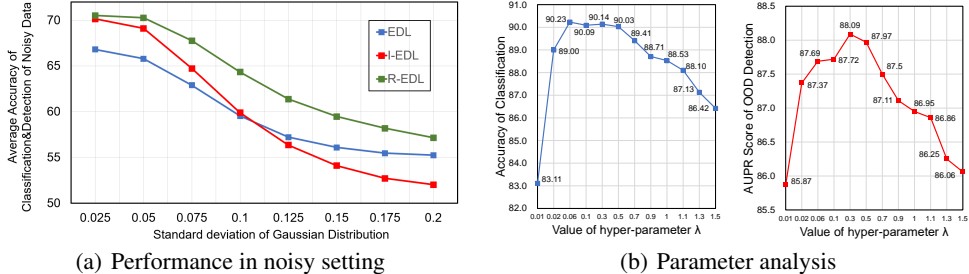

(a) Performance in noisy setting  (b) Parameter analysis

Figure 1: (a) The performance trends of EDL, $\mathcal{I}$-EDL, and R-EDL, measured by the average of the classification accuracy and the AUPR score of OOD detection, across varying levels of Gaussian noise. (b) Parameter analysis of the hyper-parameter $\lambda$, evaluated by accuracy of classification on CIFAR-10 and AUPR score of OOD detection against CIFAR-100, respectively.

Table 3: Ablation study on the classical setting and the few-shot setting, with respect to the relaxations about treating $\lambda$ as a hyper-parameter and deprecating the regularization term $\mathcal{L}_{var}$.

| Method | Classical setting | | | | Few-shot setting (10-way 5-shot) | | |
|---|---|---|---|---|---|---|---|
| | CIFAR-10 | | $\rightarrow$ SVHN | $\rightarrow$ CIFAR-100 | mini-ImageNet | | $\rightarrow$ CUB |
| | Cls | Conf | OOD Detect | OOD Detect | Cls | Conf | OOD Detect |
| EDL | 83.55±0.64 | 97.83±0.17 | 79.12±3.69 | 84.18±0.74 | 62.50±0.08 | 84.35±0.06 | 76.28±0.10 |
| $\mathcal{I}$-EDL | 89.20±0.32 | 98.63±0.11 | 82.96±2.17 | 84.84±0.64 | 67.89±0.06 | 85.52±0.05 | 82.29±0.10 |
| R-EDL w/ $\lambda=1$ | 88.53±0.38 | 98.78±0.05 | 83.24±1.25 | 86.95±0.37 | **70.77±0.06** | 85.82±0.05 | 82.89±0.09 |
| R-EDL w/ $\mathcal{L}_{var}$ | 90.03±0.15 | 98.96±0.07 | 84.04±1.66 | 87.59±0.39 | 69.97±0.05 | 85.58±0.05 | 82.25±0.09 |
| R-EDL(Ours) | **90.09±0.30** | **98.98±0.05** | **85.00±1.22** | **87.73±0.31** | 70.51±0.05 | **86.26±0.05** | **83.37±0.09** |

relaxing just one setting yields improvements of 4.12% and 4.92% respectively. Moreover, when both settings are relaxed, the performance of R-EDL improves by 5.88%. Thus, we conclude that both relaxations are effective and their joint application yields a further optimized performance.

Moreover, we further investigate the effect of the hyper-parameter $\lambda$. Fig. 1(b) demonstrates the trend of variation in classification accuracy on CIFAR-10 and the AUPR score for OOD detection on CIFAR-100 as the hyper-parameter $\lambda$ varies from 0.01 to 1.5. In this setting, $\lambda$ is ultimately established at 0.1, selected from the range [0.1:0.1:1.0] based on the best classification accuracy on the validation set. More results and analysis can be found in Appendix D.5.

Due to space limitation, please refer to Appendix D.4 for results of **Video-modality Setting**, and Appendix D.6 for **Visualization of Uncertainty Distributions** with different metrics.

# 6 CONCLUSION

**Summary.** We propose Relaxed-EDL, a generalized version of EDL, which relaxes two traditionally adopted nonessential settings in the model construction and optimization stages. Our analysis reveals two key findings: (1) A commonly ignored parameter termed prior weight governs the balance between leveraging the proportion of evidence and its magnitude in deriving predictive scores; (2) A variance-minimized regularization term adopted by the traditional EDL method encourages the Dirichlet PDF to approach a Dirac delta function, thereby heightening the risk of model overconfidence. Based on the findings, R-EDL treats the prior weight as an adjustable hyper-parameter instead of fixing it to the class number, and directly optimizes the expectation of the Dirichlet PDF provided to deprecate the variance-minimized regularization term. Comprehensive experimental evaluations underscore the efficacy of our proposed methodology.

**Deficiencies and Future directions.** This paper can be extended along two directions below. (1) Although the crucial role of the prior weight parameter in balancing the trade-off between leveraging the proportion and the magnitude of collected evidence has been elucidated, the underlying mechanism dictating its optimal value is a topic worth further investigation. (2) The optimization objective of R-EDL can be interpreted as an optimization of the expected value of the constructed Dirichlet PDF. While this approach is principled and effective, it is somewhat coarse. Future work could explore optimization goals considering other statistical properties of Dirichlet PDFs.

## ACKNOWLEDGMENTS

This work was supported in part by the National Natural Science Foundation of China under Grants 62036012, 62236008, U21B2044, 62102415, 62072286, and 62106262.

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

## A  PROOF AND DERIVATION

This section provides the proof of Theorem 1 and the derivation of optimization objectives of EDL.

### A.1  PROOF OF THEOREM 1

**Theorem 1 (Bijection between subjective opinions and Dirichlet PDFs).** Let $X$ be a random variable defined in domain $\mathbb{X}$, and $\boldsymbol{\omega}_X = (\boldsymbol{b}_X, u_X, \boldsymbol{a}_X)$ be a subjective opinion. $\boldsymbol{p}_X$ is a probability distribution over $\mathbb{X}$, and a Dirichlet PDF with the concentration parameter $\boldsymbol{\alpha}_X$ is denoted by $\mathrm{Dir}(\boldsymbol{p}_X, \boldsymbol{\alpha}_X)$, where $\boldsymbol{\alpha}_X(x) \geq 0$, and $\boldsymbol{p}_X(x) \neq 0$ if $\boldsymbol{\alpha}_X(x) < 1$. Then, given the base rate $\boldsymbol{a}_X$, there exists a bijection $F$ between the opinion $\boldsymbol{\omega}_X$ and the Dirichlet PDF $\mathrm{Dir}(\boldsymbol{p}_X, \boldsymbol{\alpha}_X)$:

$$F : \boldsymbol{\omega}_X = (\boldsymbol{b}_X, u_X, \boldsymbol{a}_X) \mapsto \mathrm{Dir}(\boldsymbol{p}_X, \boldsymbol{\alpha}_X) = \frac{\Gamma\left(\sum_{x \in \mathbb{X}} \boldsymbol{\alpha}_X(x)\right)}{\prod_{x \in \mathbb{X}} \Gamma(\boldsymbol{\alpha}_X(x))} \prod_{x \in \mathbb{X}} \boldsymbol{p}_X(x)^{\boldsymbol{\alpha}_X(x) - 1}, \quad (14)$$

where $\Gamma$ denotes the Gamma function, $\boldsymbol{\alpha}_X$ satisfies the following identity that

$$\boldsymbol{\alpha}_X(x) = \frac{\boldsymbol{b}_X(x) W}{u_X} + \boldsymbol{a}_X(x) W, \quad (15)$$

$W \in \mathbb{R}_+$ is a given scalar representing a non-informative prior weight.

*Proof.* The proof of the bijection will be performed in two steps. First, we will prove a Dirichlet distribution $\mathrm{Dir}(\boldsymbol{p}_X, \boldsymbol{\alpha}_X)$ is uniquely specified by its parameters $\alpha_X$, aka there exists a bijective mapping between $\mathrm{Dir}(\boldsymbol{p}_X, \boldsymbol{\alpha}_X)$ and $\boldsymbol{\alpha}_X$. Then, we will prove the bijection between the Dirichlet parameters $\boldsymbol{\alpha}_X$ and the subjective opinion $\boldsymbol{\omega}_X$. Therefore, the bijection between $\boldsymbol{\omega}_X$ and $\mathrm{Dir}(\boldsymbol{p}_X, \boldsymbol{\alpha}_X)$ can be established due to the transitivity of bijection.

Step 1: To prove the mapping $F_1 : \boldsymbol{\alpha}_X \mapsto \mathrm{Dir}(\boldsymbol{p}_X, \boldsymbol{\alpha}_X)$ is bijective, we will prove it is both injective and surjective. The surjective property is obvious due to the mapping form. We use proof by contradiction to verify the injectivity as follows.

Assuming that there exists two Dirichlet distributions over the random variable $X$, which are parameterized by two different concentration parameter vectors $\boldsymbol{\alpha}_X$ and $\tilde{\boldsymbol{\alpha}}_X$ respectively, sharing exactly the same probability density function, i.e., $\exists x \in \mathbb{X}, \boldsymbol{\alpha}_X(x) \neq \tilde{\boldsymbol{\alpha}}_X(x)$, and $\forall x \in \mathbb{X}$ and $\forall \boldsymbol{p}_X \in \mathcal{S}_{|\mathbb{X}|}$,

$$\frac{\Gamma\left(\sum_{x \in \mathbb{X}} \boldsymbol{\alpha}_X(x)\right)}{\prod_{x \in \mathbb{X}} \Gamma(\boldsymbol{\alpha}_X(x))} \prod_{x \in \mathbb{X}} \boldsymbol{p}_X(x)^{\boldsymbol{\alpha}_X(x) - 1} = \frac{\Gamma\left(\sum_{x \in \mathbb{X}} \tilde{\boldsymbol{\alpha}}_X(x)\right)}{\prod_{x \in \mathbb{X}} \Gamma(\tilde{\boldsymbol{\alpha}}_X(x))} \prod_{x \in \mathbb{X}} \boldsymbol{p}_X(x)^{\tilde{\boldsymbol{\alpha}}_X(x) - 1}, \quad (16)$$

where $\mathcal{S}_{|\mathbb{X}|}$ is a $|\mathbb{X}|$-dimensional unit simplex. Taking the logarithm of both sides, we have

$$-\log(B(\boldsymbol{\alpha}_X(x))) + \sum_{x \in \mathbb{X}} (\boldsymbol{\alpha}_X(x) - 1) \log(\boldsymbol{p}_X(x)) = -\log(B(\tilde{\boldsymbol{\alpha}}_X(x))) + \sum_{x \in \mathbb{X}} (\tilde{\boldsymbol{\alpha}}_X(x) - 1) \log(\boldsymbol{p}_X(x)),$$
$$(17)$$

where $B$ denotes a $|\mathbb{X}|$-dimensional beta function. Therefore, we have

$$\sum_{x \in \mathbb{X}} (\boldsymbol{\alpha}_X(x) - \tilde{\boldsymbol{\alpha}}_X(x)) \log(\boldsymbol{p}_X(x)) = \log\left(\frac{B(\boldsymbol{\alpha}_X(x))}{B(\tilde{\boldsymbol{\alpha}}_X(x))}\right), \quad \forall \boldsymbol{p}_X \in \mathcal{S}_{|\mathbb{X}|}. \quad (18)$$

Since the above equation holds for any probability distribution $\boldsymbol{p}_X$, we have

$$\sum_{x \in \mathbb{X}} (\boldsymbol{\alpha}_X(x) - \tilde{\boldsymbol{\alpha}}_X(x)) \log(\boldsymbol{p}_X(x) - \boldsymbol{p}'_X(x)) = 0, \quad \forall \boldsymbol{p}_X, \boldsymbol{p}'_X \in \mathcal{S}_{|\mathbb{X}|}. \quad (19)$$

The above equation can be regarded as a homogenous linear equation with $\boldsymbol{\alpha}_X(x) - \tilde{\boldsymbol{\alpha}}_X(x)$ as variables and $\log(\boldsymbol{p}_X(x) - \boldsymbol{p}'_X(x))$ as parameters. Due to the arbitrariness of $\boldsymbol{p}_X$ and $\boldsymbol{p}'_X$, and the property of homogeneous systems of linear equations, we know that Eqn. 19 only has a particular solution, i.e., $\boldsymbol{\alpha}_X(x) - \tilde{\boldsymbol{\alpha}}_X(x) = 0$ for any $x \in \mathbb{X}$, which violates our assumption.

Therefore, $F_1$ is both injective and surjective, thus bijective.

Step 2: To prove the bijection between $\boldsymbol{\omega}_X$ and $\boldsymbol{\alpha}_X$, we also need to prove the mapping $F_2 : \boldsymbol{\omega}_X \mapsto \boldsymbol{\alpha}_X$ is both injective and surjective. Since the base rate $\boldsymbol{a}_X$ and the non-informative prior weight $W$ in Eqn. 15 are given, $F_2$ can be simplified to $(\boldsymbol{b}_X, u_X) \mapsto \boldsymbol{\alpha}_X$ with the following formulation:

$$\boldsymbol{\alpha}_X(x) = \frac{\boldsymbol{b}_X(x)}{u_X}, \quad \forall x \in \mathbb{X}. \quad (20)$$

First, we use proof by contradiction to verify the injection. Assuming that there exists two different sets of belief mass and uncertainty mass which corresponds to the same set of Dirichlet concentration parameters, aka there exists $(\boldsymbol{b}_X, u_X), (\tilde{\boldsymbol{b}}_X, \tilde{u}_X), \boldsymbol{\alpha}_X$, which satisfies

$$\boldsymbol{\alpha}_X(x) = \frac{\boldsymbol{b}_X(x)}{u_X} = \frac{\tilde{\boldsymbol{b}}_X(x)}{\tilde{u}_X}, \quad \forall x \in \mathbb{X}, \tag{21}$$

and $\exists x \in \mathbb{X}, \boldsymbol{b}_X(x) \neq \tilde{\boldsymbol{b}}_X(x)$, or $u_X \neq \tilde{u}_X$. We take the summation of Eqn. 21 across all possible values of $x \in \mathbb{X}$ and utilize the additivity requirement $\sum_{x \in \mathbb{X}} \boldsymbol{b}_X(x) + u_X = 1$, then we will have

$$\sum_{x \in \mathbb{X}} \boldsymbol{\alpha}_X(x) = \frac{1 - u_X}{u_X} = \frac{1 - \tilde{u}_X}{\tilde{u}_X}. \tag{22}$$

Thus we reach $u_X = \tilde{u}_X$ and after using the relationship in Eqn. 21, we will have $\boldsymbol{b}_X(x) = \tilde{\boldsymbol{b}}_X(x)$, $\forall x \in \mathbb{X}$. Thereafter, our assumption is violated and thus $F_2$ is injective.

Second, we prove $F_2$ is surjective, aka for any Dirichlet parameter set $\boldsymbol{\alpha}_X$, there exists a set of $(\boldsymbol{b}_X, u_X)$ satisfying Eqn. 20. By summing Eqn. 20 over all values of $x \in \mathbb{X}$, we have

$$S_X = \frac{1 - u_X}{u_X}, \tag{23}$$

where $S_X = \sum_{x \in \mathbb{X}} \boldsymbol{\alpha}_X(x)$. By reorganization and substituting $u_X$ into Eqn. 20, we have

$$u_X = \frac{1}{S_X + 1}, \quad \boldsymbol{b}_X(x) = \frac{\boldsymbol{\alpha}_X(x)}{S_X + 1}, \tag{24}$$

which satisfy all the requirements. Therefore, the mapping $F_2$ is surjective.

Finally, since $F_1$ and $F_2$ are both bijective, $F = F_1 \circ F_2$ is also bijective. $\qquad\square$

Moreover, in cases of no prior information available, we generally set the base rate $\boldsymbol{a}_X(x)$ as uniform distribution, i.e., $\boldsymbol{a}_X(x) = \frac{1}{|\mathbb{X}|}, \forall x \in \mathbb{X}$, and Eqn. 15 can be reorganized as

$$\boldsymbol{\alpha}_X(x) = \left( \frac{\boldsymbol{b}_X(x)}{u_X} + \frac{1}{|\mathbb{X}|} \right) W, \quad \forall x \in \mathbb{X}, \tag{25}$$

or equivalently as

$$\boldsymbol{b}_X(x) = \frac{\boldsymbol{\alpha}_X(x) - W/|\mathbb{X}|}{\sum_{x' \in \mathbb{X}} \boldsymbol{\alpha}_X(x')}, \quad u_X = \frac{W}{\sum_{x \in \mathbb{X}} \boldsymbol{\alpha}_X(x)}, \quad \forall x \in \mathbb{X}, \tag{26}$$

by utilizing the additivity requirement $\sum_{x \in \mathbb{X}} \boldsymbol{b}_X(x) + u_X = 1$.

Besides, it is noteworthy that comprehensive elaborations on the concepts within the Subjective Logic theory are available in Jøsang (2001; 2016).

## A.2 Derivation of Optimization Objectives in EDL

As aforementioned in section 3.1, to perform model optimization, EDL integrates the conventional MSE loss function over the class probability $\boldsymbol{p}_X$ which is assumed to follow the Dirichlet PDF specified in the bijection, thus derives the optimization objective as

$$\begin{aligned}
\mathcal{L}_{edl} &= \sum_{(\boldsymbol{z}, \boldsymbol{y}) \in \mathcal{D}} \mathbb{E}_{\boldsymbol{p}_X \sim \text{Dir}(\boldsymbol{p}_X, \boldsymbol{\alpha}_X)} \left[ \|\boldsymbol{y} - \boldsymbol{p}_X\|_2^2 \right] \\
&= \sum_{(\boldsymbol{z}, \boldsymbol{y}) \in \mathcal{D}} \mathbb{E}_{\boldsymbol{p}_X \sim \text{Dir}(\boldsymbol{p}_X, \boldsymbol{\alpha}_X)} \sum_{x \in \mathbb{X}} \left( \boldsymbol{y}[x]^2 - 2\boldsymbol{y}[x]\boldsymbol{p}_X(x) + \boldsymbol{p}_X(x)^2 \right) \\
&= \sum_{(\boldsymbol{z}, \boldsymbol{y}) \in \mathcal{D}} \sum_{x \in \mathbb{X}} \left( \boldsymbol{y}[x]^2 - 2\boldsymbol{y}[x]\mathbb{E}_{\boldsymbol{p}_X \sim \text{Dir}(\boldsymbol{p}_X, \boldsymbol{\alpha}_X)}[\boldsymbol{p}_X(x)] + \mathbb{E}_{\boldsymbol{p}_X \sim \text{Dir}(\boldsymbol{p}_X, \boldsymbol{\alpha}_X)} \left[ \boldsymbol{p}_X(x)^2 \right] \right).
\end{aligned} \tag{27}$$

Using the identity $\mathbb{E}[x^2] = \mathbb{E}[x]^2 + \text{Var}[x]$, we know that

$$\mathcal{L}_{edl} = \sum_{(\boldsymbol{z}, \boldsymbol{y}) \in \mathcal{D}} \sum_{x \in \mathbb{X}} \left( \boldsymbol{y}[x] - \mathbb{E}_{\boldsymbol{p}_X \sim \text{Dir}(\boldsymbol{p}_X, \boldsymbol{\alpha}_X)}[\boldsymbol{p}_X(x)] \right)^2 + \text{Var}_{\boldsymbol{p}_X \sim \text{Dir}(\boldsymbol{p}_X, \boldsymbol{\alpha}_X)}[\boldsymbol{p}_X(x)]. \tag{28}$$

Since the Dirichlet distribution has the following properties:

$$\mathbb{E}[\boldsymbol{p}_X(x)] = \frac{\boldsymbol{\alpha}_X(x)}{S_X}, \quad \text{Var}[\boldsymbol{p}_X(x)] = \frac{\boldsymbol{\alpha}_X(x)(S_X - \boldsymbol{\alpha}_X(x))}{S_X^2(S_X + 1)}, \tag{29}$$

where $S_X = \sum_{i=1}^{C} \boldsymbol{\alpha}_X(x)$, we can explicitly express $\mathcal{L}_{edl}$ by $\boldsymbol{\alpha}_X(x)$ and $S_X$ as

$$\mathcal{L}_{edl} = \sum_{(\boldsymbol{z},\boldsymbol{y}) \in \mathcal{D}} \sum_{x \in \mathbb{X}} \left( \boldsymbol{y}[x] - \frac{\boldsymbol{\alpha}_X(x)}{S_X} \right)^2 + \frac{\boldsymbol{\alpha}_X(x)(S_X - \boldsymbol{\alpha}_X(x))}{S_X^2(S_X + 1)}. \tag{30}$$

Furthermore, EDL introduces an auxiliary regularization term to suppress the evidence of non-target classes by minizing the Kullback-Leibler (KL) divergence between a modified Dirichlet distribution and a uniform distribution. This regularization term has demonstrated promising empirical results and has been elucidated by Deng et al. (2023) using the PAC-Bayesian theory (McAllester, 1998). Specifically, the regularization term has the following form:

$$\mathcal{L}_{kl} = \frac{1}{|\mathcal{D}|} \sum_{(\boldsymbol{z},\boldsymbol{y}) \in \mathcal{D}} \text{KL}\left(\text{Dir}(\boldsymbol{p}_X, \tilde{\boldsymbol{\alpha}}_X), \text{Dir}(\boldsymbol{p}_X, \mathbf{1})\right)$$

$$= \frac{1}{|\mathcal{D}|} \sum_{(\boldsymbol{z},\boldsymbol{y}) \in \mathcal{D}} \left( \log \frac{\Gamma(S_X)}{\Gamma(C) \prod_{x \in \mathbb{X}} \Gamma(\tilde{\boldsymbol{\alpha}}_X(x))} + \sum_{x \in \mathbb{X}} (\tilde{\boldsymbol{\alpha}}_X(x) - 1)(\psi(\tilde{\boldsymbol{\alpha}}_X(x)) - \psi(S_X)) \right), \tag{31}$$

where $\Gamma$ denotes the Gamma function, and $\tilde{\boldsymbol{\alpha}}_X = \boldsymbol{y} + (\mathbf{1} - \boldsymbol{y}) \odot \boldsymbol{\alpha}_X$ represents a modified Dirichlet parameter vector whose value of the target class has been set to 1.

## B    DERIVATION FOR UNCERTAINTY MEASURES

This section provides the derivation of several uncertainty measures, including expected entropy, mutual information, and differential entropy, of Dirichlet-based uncertainty models. The following content is adapted from the Appendix of Malinin & Gales (2018) and Deng et al. (2023).

### B.1    EXPECTED ENTROPY

Let $X$ be a random variable defined in $\mathbb{X}$, where $\mathbb{X}$ is a domain consisting of multiple mutually disjoint values. Let $\boldsymbol{p}$ be a probability distribution over $\mathbb{X}$, and let $\text{Dir}(\boldsymbol{p}, \boldsymbol{\alpha})$ be a Dirichlet distribution parameterized by the concentration parameter vector $\boldsymbol{\alpha}$. If $X$ represents the category index of an input sample, $x \in \mathbb{X} = \{1, ..., C\}$ denotes the value of $X$, satisfying $p(X = x) = \boldsymbol{p}(x)$, then the expected entropy of the random variable $X$ over the Dirichlet distribution $\text{Dir}(\boldsymbol{p}, \boldsymbol{\alpha})$ can be derived as follows:

$$\mathbb{E}_{\boldsymbol{p} \sim \text{Dir}(\boldsymbol{p}, \boldsymbol{\alpha})}[\mathcal{H}[\boldsymbol{p}(x)]] = \int_{\boldsymbol{p} \in \mathcal{N}_C} \text{Dir}(\boldsymbol{p}, \boldsymbol{\alpha}) \left( -\sum_{x \in \mathbb{X}} \boldsymbol{p}(x) \ln \boldsymbol{p}(x) \right) d\boldsymbol{p}$$

$$= -\sum_{x \in \mathbb{X}} \int_{\boldsymbol{p} \in \mathcal{N}_C} \text{Dir}(\boldsymbol{p}, \boldsymbol{\alpha}) (-\boldsymbol{p}(x) \ln \boldsymbol{p}(x)) d\boldsymbol{p}$$

$$= -\sum_{x \in \mathbb{X}} \int_{\boldsymbol{p} \in \mathcal{N}_C} \frac{\Gamma(S)}{\prod_{x' \in \mathbb{X}} \Gamma(\boldsymbol{\alpha}(x'))} \prod_{x' \in \mathbb{X}} \boldsymbol{p}(x')^{\boldsymbol{\alpha}(x')-1} (-\boldsymbol{p}(x) \ln \boldsymbol{p}(x)) d\boldsymbol{p}$$

$$= -\sum_{x \in \mathbb{X}} \int_{\boldsymbol{p} \in \mathcal{N}_C} \frac{\boldsymbol{\alpha}(x)}{S} \frac{\Gamma(S)}{\Gamma(\boldsymbol{\alpha}(x)+1) \prod_{x' \neq x} \Gamma(\boldsymbol{\alpha}(x'))} \prod_{x' \neq x} \boldsymbol{p}(x')^{\boldsymbol{\alpha}(x')-1} \boldsymbol{p}(x)^{\boldsymbol{\alpha}(x)} \ln \boldsymbol{p}(x) d\boldsymbol{p} \tag{32}$$

$$= -\sum_{x \in \mathbb{X}} \frac{\boldsymbol{\alpha}(x)}{S} \int_{\boldsymbol{p} \in \mathcal{N}_C} \mathbb{E}_{\boldsymbol{p} \sim \text{Dir}(\boldsymbol{p}, \boldsymbol{\alpha}+\mathbf{1}_x)}[\ln \boldsymbol{p}(x)] d\boldsymbol{p}$$

$$= -\sum_{x \in \mathbb{X}} \frac{\boldsymbol{\alpha}(x)}{S} (\psi(\boldsymbol{\alpha}(x)+1) - \psi(S+1)),$$

where $S = \sum_{x \in \mathbb{X}} \boldsymbol{\alpha}(x)$, $\mathcal{N}_C$ is a $C$-dimensional unit simplex, $\psi$ denotes the digamma function, and $\mathbf{1}_x$ denotes a one-hot vector with the $x$-th element being set to 1 and other elements being set to 0. The last third equation comes from the property of Gamma function that $\Gamma(n) = (n-1)!$. In some literature, the expected entropy is used to measure the *data uncertainty*.

## B.2   MUTUAL INFORMATION

In the Dirichlet-based uncertainty methods, the mutual information between the labels $\boldsymbol{y}$ and the class probability $\boldsymbol{p}$, which can be regarded as the difference between the total amount of uncertainty and the data uncertainty, can be approximately computed as:

$$
\underbrace{I[\boldsymbol{y}, \boldsymbol{p}]}_{\text{Distributional Uncertainty}} \approx \underbrace{\mathcal{H}\left[\mathbb{E}_{\boldsymbol{p} \sim \text{Dir}(\boldsymbol{p}, \boldsymbol{\alpha})}[\boldsymbol{p}(x)]\right]}_{\text{Total Uncertainty}} - \underbrace{\mathbb{E}_{\boldsymbol{p} \sim \text{Dir}(\boldsymbol{p}, \boldsymbol{\alpha})}\left[\mathcal{H}[\boldsymbol{p}(x)]\right]}_{\text{Expected Data Uncertainty}}
$$

$$
= -\sum_{x \in \mathbb{X}} \frac{\boldsymbol{\alpha}(x)}{S} \ln \frac{\boldsymbol{\alpha}(x)}{S} + \sum_{x \in \mathbb{X}} \frac{\boldsymbol{\alpha}(x)}{S} \left(\psi(\boldsymbol{\alpha}(x) + 1) - \psi(S + 1)\right) \qquad (33)
$$

$$
= -\sum_{x \in \mathbb{X}} \frac{\boldsymbol{\alpha}(x)}{S} \left(\ln \frac{\boldsymbol{\alpha}(x)}{S} - \psi(\boldsymbol{\alpha}(x) + 1) + \psi(S + 1)\right).
$$

The calculation of the expected data uncertainty utilizes the result of Eqn. 32. The mutual information is often used to measure the *distributional uncertainty*.

## B.3   DIFFERENTIAL ENTROPY

The derivation of the differential entropy of the Dirichlet distribution is given by:

$$
\mathcal{H}[\text{Dir}(\boldsymbol{p}, \boldsymbol{\alpha})] = -\int_{\boldsymbol{p} \in \mathcal{N}_C} \text{Dir}(\boldsymbol{p}, \boldsymbol{\alpha}) \ln \text{Dir}(\boldsymbol{p}, \boldsymbol{\alpha}) d\boldsymbol{p}
$$

$$
= -\int_{\boldsymbol{p} \in \mathcal{N}_C} \text{Dir}(\boldsymbol{p}, \boldsymbol{\alpha}) \left(\ln \Gamma(S) - \sum_{x \in \mathbb{X}} \Gamma(\boldsymbol{\alpha}(x)) + \sum_{x \in \mathbb{X}} (\boldsymbol{\alpha}(x) - 1) \ln \boldsymbol{p}(x)\right) d\boldsymbol{p}
$$

$$
= \sum_{x \in \mathbb{X}} \ln \Gamma(\boldsymbol{\alpha}(x)) - \ln \Gamma(S) - \sum_{x \in \mathbb{X}} (\boldsymbol{\alpha}(x) - 1) \mathbb{E}_{\boldsymbol{p} \sim \text{Dir}(\boldsymbol{p}, \boldsymbol{\alpha})}[\ln \boldsymbol{p}(x)]
$$

$$
= \sum_{x \in \mathbb{X}} \ln \Gamma(\boldsymbol{\alpha}(x)) - \ln \Gamma(S) - \sum_{x \in \mathbb{X}} (\boldsymbol{\alpha}(x) - 1)(\psi(\boldsymbol{\alpha}(x) - \psi(S))).
$$

$$
(34)
$$

Differential entropy is also a prevalent measure of *distributional uncertainty*. A lower entropy indicates that the model yields a sharper distribution, whereas a higher value signifies a more uniform Dirichlet distribution.

## C   EXPERIMENTAL SETTINGS

### C.1   DATASETS

Following Deng et al. (2023), we conduct experiments on the following groups of image classification dataset: (1) **MNIST** (LeCun, 1998), **FMNIST** (Xiao et al., 2017), **KMNIST** (Clanuwat et al., 2018); (2) **CIFAR-10** (Krizhevsky et al., 2009), **SVHN** (Netzer et al., 2018), **CIFAR-100** (Krizhevsky et al., 2009); (3) **mini-ImageNet** (Vinyals et al., 2016), **CUB** (Wah et al., 2011). Within each group, we designate the first dataset as in-distribution training data, while utilizing the subsequent ones as OOD data. Moreover, to evaluate the effectiveness of our method on video-modality data, we also conduct an open-set action recognition experiment by taking **UCF-101** (Soomro et al., 2012) as ID data and **HMDB-51** (Kuehne et al., 2011) and **MiT-v2** (Monfort et al., 2021) as OOD data following Bao et al. (2021). Below are the detailed introductions:

The **MNIST** (LeCun, 1998) database consists of handwritten digits ranging from 0 to 9. Specifically, MNIST contains 60,000 training images and 10,000 testing images, which have been normalized to fit into $28 \times 28$ pixel bounding boxes. We use the proportion of [0.8, 0.2] to partition the training samples into training and validation sets.

FashionMNIST (**FMNIST**) (Xiao et al., 2017) is a dataset designed as a more challenging replacement for MNIST. Created by Zalando Research, FMNIST features grayscale images of various clothing items such as shirts, trousers, sneakers, and bags. The dataset is structured similarly to MNIST, containing 60,000 training images and 10,000 testing images, each of which is $28 \times 28$ pixels in size. We use FMNIST as OOD data when training models on MNIST.

Kuzushiji-MNIST (**KMNIST**) (Clanuwat et al., 2018) is another drop-in replacement for MNIST, consisting of a training set with 60,000 handwritten Kuzushiji (cursive Japanese) Hiragana characters and a testing set comprising 10,000 ones. Similar to MNIST, the handwritten characters have been processed to fit into $28 \times 28$ pixel resolution grayscale images. We also use KMNIST as OOD data when using MNIST as ID data.

**CIFAR-10** (Krizhevsky et al., 2009) comprises 60,000 $32 \times 32$ color distributed across 10 distinct classes such as airplanes, birds, cats, ships, and more, with each class containing 6,000 images. Among them, 50,000 are designated for training and the remaining 10,000 for testing. We partition the training images into training and validation sets using a split ratio of [0.95, 0.05].

Street View House Numbers (**SVHN**) (Netzer et al., 2018) dataset consists of digit images of house numbers from Google Street View. Specifically, it contains 73257 digits for training and 26032 digits for testing. We use SVHN as OOD data when training models on CIFAR10.

**CIFAR-100** (Krizhevsky et al., 2009) is just like the CIFAR-10, except it has 100 classes containing 600 images each. There are 500 training images and 100 testing images per class. We use CIFAR-100 as OOD data when using CIFAR-10 as ID data.

**mini-ImageNet** (Vinyals et al., 2016) is designed for few-shot learning evaluation. mini-ImageNet comprises 50,000 $84 \times 84$ color images for training and 10,000 ones for testing, evenly distributed across 100 classes, and these 100 classes are subdivided into sets of 64, 16, and 20 for meta-training, meta-validation, and meta-testing tasks, respectively.

The Caltech-UCSD Birds (**CUB**) (Wah et al., 2011) dataset contains 11,788 images of 200 subcategories belonging to birds, 5,994 for training and 5,794 for testing. We use CUB as OOD data when using mini-ImageNet as ID data in the few-shot setting.

**UCF-101** (Soomro et al., 2012) is an action recognition data set of realistic action videos, collected from YouTube. Specifically, UCF-101 contains 13320 videos distributed across 101 action categories. For experiments of video-modality setting, we train models on UCF-101 training split and take its testing set as known samples in inference. Following Bao et al. (2021), despite there exists a few overlapping classes between UCF-101 and the OOD datasets, HMDB-51 and MiT-v2, we do not manually clean the data for standardizing the evaluation.

**HMDB-51** (Kuehne et al., 2011) is collected mostly from movies, and a small proportion from Prelinger archive, YouTube and Google videos. Specifically, HMDB-51 contains 6,849 clips of 51 action categories, each containing a minimum of 101 clips. We use its testing set as unknown samples in the video-modality setting.

Multi-Moments in Time (**MiT-v2**) (Monfort et al., 2021) has 305 classes and its testing split contains 30,500 video samples. We also use its testing set as unknown samples in the video-modality setting.

## C.2 IMPLEMENTATION DETAILS

**Classical setting.** In alignment with Deng et al. (2023), a ConvNet with three convolutional and three dense layers is employed for MNIST, while VGG16 (Simonyan & Zisserman, 2014) serves as the backbone network for CIFAR-10. As Table 4 shows, FMNIST and KMNIST are utilized as OOD data for MNIST, while SVHN and CIFAR-100 are used for CIFAR-10. We use Softplus as the activation function to keep the collected evidence non-negative. The Adam optimizer is employed with a learning rate of $1 \times 10^{-3}$, decaying by 0.1 every 15 epochs for MNIST, and a learning rate of $1 \times 10^{-4}$ for CIFAR-10. The hyper-parameter $\lambda$ is set to 0.1, which is selected from the range [0.1:0.1:1.0] based on the optimal classification accuracy on the validation set. The batch size is set to 64, and the maximum training epoch is set to 60 and 200 for MNIST and CIFAR-10, respectively. Reported results are averaged over 5 runs.

Besides, for the baseline methods which require OOD data in the training phase, i.e., KL-PN and RKL-PN, uniform noise instead of actual OOD test data is used as OOD training data to ensure a fair comparison as previous works did (Charpentier et al., 2020; Deng et al., 2023).

Table 4: Implementation details of experiments in the classical setting.

| ID dataset | OOD dataset | Optimizer | Learning rate | (decay,step) for lr | Max Epoch | $\lambda$ |
|---|---|---|---|---|---|---|
| MNIST | FMNIST & KMNIST | Adam | $1 \times 10^{-3}$ | (0.1, 15) | 60 | 0.1 |
| CIFAR-10 | SVHN & CIFAR-100 | Adam | $1 \times 10^{-4}$ | - | 200 | 0.1 |

**Few-shot setting.** Following Deng et al. (2023), we adopt a pre-trained WideResNet-28-10 network from Yang et al. (2021) to extract features and train a single dense layer for experiments under a challenging few-shot setting on the mini-ImageNet dataset, with the testing set of CUB as OOD data. We employ the $N$-way $K$-shot setting, with $N \in \{5, 10\}$ and $K \in \{1, 5, 20\}$. Each few-shot episode comprises $N$ random classes and $K$ random samples per class for training, $\min(15, K)$ query samples per class from mini-ImageNet for classification and confidence estimation, and an equivalent number of query samples from the CUB dataset for OOD detection. Reported results are averaged over 10,000 episodes. Note that in the few-shot setting, we perform setting relaxations on $\mathcal{I}$-EDL to achieve stronger performances. Softplus is used as the activation function to keep evidence non-negative. The LBFGS optimizer is employed with the default learning rate 1.0 for 100 epochs. The hyper-parameter $\lambda$ is also selected on the meta-validation set, as shown in Table 5.

Table 5: List of the hyper-parameter $\lambda$ of experiments in the few-shot setting.

| Setting | 5-Way 1-Shot | 5-Way 5-Shot | 5-Way 20-Shot | 10-Way 1-Shot | 10-Way 5-Shot | 10-Way 20-Shot |
|---|---|---|---|---|---|---|
| $\lambda$ | 0.7 | 0.2 | 0.3 | 0.8 | 0.6 | 0.7 |

**Noisy setting**. Noisy samples are generated by adding zero-mean Gaussian noises with standard deviations of [0.025:0.025:0.200] to the testing samples of CIFAR-10. The hyper-parameter $\lambda$ is set to 0.3, which is selected by the best AUPR score of OOD detection on the clean validation set of CIFAR-10 against the noisy validation set with zero-mean 0.1-SD Gaussian noise.

**Video-modality setting.** Following Bao et al. (2021), we explore the open-set action recognition task on UCF-101 with I3D as the backbone network. The HMDB-51 and MiT-v2 are used as sources of unknown samples. The hyper-parameter $\lambda$ is set to 0.8, and the batch size is set to 8.

**Other details.** Our model is implemented with Python 3.8 and PyTorch 1.12. All experiments are conducted on NVIDIA RTX 3090 GPUs. Source codes are provided in the supplementary material.

# D  ADDITIONAL RESULTS

## D.1  CLASSICAL SETTING

In Table 6 and Table 7, we provide the AUPR and AUROC scores of OOD detection in the classical setting, measured by MP (Max projected probability), UM (Uncertainty Mass), DE (Differential Entropy), and MI (Mutual Information), respectively. Table 8 compares EDL-related works with the temperature scaling method (Guo et al., 2017) in the classical setting, including results evaluated by the Expected Calibration Error (ECE) with 15 bins and the Brier score. Although temperature scaling achieves impressive results when evaluated by the ECE metric, there still exists a performance gap with our method on OOD detection ability.

Besides, we believe that employing the AUPR scores for evaluation purposes aligns more closely with our objectives than using ECE or Brier score. As delineated in Section 5.2, our primary criterion for assessing confidence estimation is the model's ability in differentiating between correctly classified and misclassified samples, as well as between ID and OOD samples based on the predicted confidence. Despite that ECE is frequently employed to assess the degree of correspondence between the model's confidence and the true correctness likelihood, a confidence distribution accompanied by a low ECE does not inherently ensure a clear distinction between correct and incorrect predictions. For instance, in a balanced two-class dataset scenario, if a binary classifier categorizes

Table 6: AUPR scores of OOD detection in the classical setting, measured by MP (Max projected probability), UM (Uncertainty Mass), DE (Differential Entropy), and MI (Mutual Information).

| Method | MNIST→KMNIST | | | | MNIST→FMNIST | | | |
| | MP | UM | DE | MI | MP | UM | DE | MI |
|---|---|---|---|---|---|---|---|---|
| EDL | 97.02±0.76 | 96.31±2.03 | 96.92±0.91 | 96.41±1.85 | 98.10±0.44 | 98.08±0.42 | 98.10±0.43 | 98.09±0.42 |
| $\mathcal{I}$-EDL | 98.34±0.24 | 98.33±0.24 | 98.34±0.24 | 98.33±0.24 | 98.89±0.28 | 98.86±0.29 | 98.89±0.29 | 98.87±0.29 |
| R-EDL(Ours) | **98.69±0.19** | **98.69±0.20** | **98.70±0.19** | **98.69±0.19** | **99.29±0.11** | **99.29±0.12** | **99.31±0.12** | **99.29±0.12** |

| Method | CIFAR-10→SVHN | | | | CIFAR-10→CIFAR-100 | | | |
| | MP | UM | DE | MI | MP | UM | DE | MI |
|---|---|---|---|---|---|---|---|---|
| EDL | 78.87±3.50 | 79.12±3.69 | 78.91±3.54 | 79.11±3.68 | 84.30±0.67 | 84.18±0.74 | 84.32±0.67 | 84.19±0.74 |
| $\mathcal{I}$-EDL | 83.26±2.44 | 82.96±2.17 | 83.31±2.47 | 83.07±2.27 | 85.35±0.69 | 84.84±0.64 | 85.40±0.69 | 84.95±0.65 |
| R-EDL(Ours) | **85.00±1.22** | **85.00±1.22** | **85.01±1.14** | **85.00±1.22** | **87.72±0.31** | **87.73±0.31** | **87.61±0.33** | **87.73±0.31** |

Table 7: AUROC scores of OOD detection in the classical setting, measured by MP (Max projected probability), UM (Uncertainty Mass), DE (Differential Entropy), and MI (Mutual Information).

| Method | MNIST→KMNIST | | | | MNIST→FMNIST | | | |
| | MP | UM | DE | MI | MP | UM | DE | MI |
|---|---|---|---|---|---|---|---|---|
| EDL | 96.59±0.59 | 96.18±1.35 | 96.49±0.80 | 96.22±1.29 | 97.39±0.57 | 97.40±0.54 | 97.48±0.53 | 97.43±0.53 |
| $\mathcal{I}$-EDL | 98.00±0.26 | 97.97±0.26 | 97.99±0.26 | 97.97±0.26 | 98.49±0.36 | 98.41±0.39 | 98.48±0.37 | 98.42±0.39 |
| R-EDL(Ours) | **98.40±0.18** | **98.39±0.18** | **98.42±0.18** | **98.40±0.18** | **98.99±0.14** | **98.98±0.14** | **99.05±0.14** | **98.98±0.14** |

| Method | CIFAR10→SVHN | | | | CIFAR10→CIFAR100 | | | |
| | MP | UM | DE | MI | MP | UM | DE | MI |
|---|---|---|---|---|---|---|---|---|
| EDL | 80.64±4.22 | 81.06±4.52 | 80.72±4.33 | 81.05±4.51 | 80.96±0.81 | 80.63±1.01 | 80.99±0.83 | 80.65±1.00 |
| $\mathcal{I}$-EDL | **87.58±2.03** | 86.79±1.35 | **87.69±2.09** | 87.01±1.52 | 83.55±0.67 | 82.15±0.50 | 83.69±0.68 | 82.44±0.50 |
| R-EDL(Ours) | 87.47±1.22 | **87.47±1.24** | 87.54±0.96 | **87.47±1.24** | **85.26±0.36** | **85.26±0.35** | **84.90±0.45** | **85.26±0.35** |

all samples into a single class with a consistent confidence output of 50%, the ECE would be zero, yet this result lacks practical significance.

## D.2 FEW-SHOT SETTING

Table 9 shows few-shot results of OOD detection measured by more uncertainty metrics. Table 10 compares our method and label smoothing in the few-shot setting. All results consistently demonstrate the superior OOD detection performance of our proposed method.

## D.3 NOISY SETTING

We also employ noisy data to assess both the robustness of classification and the OOD detection capability of our method with the interference of noise. Following Deng et al. (2023), we generate the noisy data by introducing zero-mean isotropic Gaussian noise to the test split of the ID dataset. Table 11 reports the classification accuracy and the AUPR scores for OOD detection across varying levels of Gaussian noise on CIFAR-10. It is essential to note that these two metrics are not mutually

Table 8: Comparison of temperature scaling method with EDL-related works in the classical setting, including results evaluated by the Expected Calibration Error (ECE) with 15 bins and the Brier score. Downward arrows (↓) indicate that lower values correspond to better performance for these metrics.

| Method | Confidence estimation | | | OOD detection | |
| | ECE ↓ | Brier score ↓ | AUPR | AURP (SVHN) | AUPR (CIFAR100) |
|---|---|---|---|---|---|
| Temperature scaling | **1.06±0.10** | 18.44±0.49 | 98.89±0.05 | 81.89±2.19 | 86.86±0.48 |
| EDL | 20.08±1.77 | 40.68±2.39 | 97.86±0.17 | 78.87±3.50 | 84.30±0.67 |
| I-EDL | 39.96±0.37 | 55.32±0.50 | 98.72±0.12 | 83.26±2.44 | 85.35±0.69 |
| R-EDL(Ours) | 3.48±0.30 | **18.14±0.51** | **98.98±0.05** | **85.00±1.22** | **87.72±0.31** |

Table 9: AUPR scores of OOD detection in the few-shot setting, measured by MP (Max projected probability), UM (Uncertainty Mass), DE (Differential Entropy), and MI (Mutual Information).

| Method | 5-Way 1-Shot | | | | 10-Way 1-Shot | | | |
| | MP | UM | DE | MI | MP | UM | DE | MI |
|---|---|---|---|---|---|---|---|---|
| EDL | 66.78±0.12 | 65.41±0.13 | 69.00±0.12 | 66.11±0.13 | 59.19±0.09 | 67.81±0.12 | 67.78±0.12 | 67.84±0.12 |
| $\mathcal{I}$-EDL | 71.79±0.12 | 74.76±0.13 | 74.04±0.13 | 74.70±0.13 | 71.60±0.10 | 71.95±0.10 | 71.57±0.10 | 71.95±0.10 |
| R-EDL(Ours) | **72.91±0.12** | **74.84±0.13** | **74.34±0.13** | **74.76±0.13** | **72.83±0.10** | **73.08±0.10** | **73.17±0.10** | **73.08±0.10** |

| Method | 5-Way 5-Shot | | | | 10-Way 5-Shot | | | |
| | MP | UM | DE | MI | MP | UM | DE | MI |
|---|---|---|---|---|---|---|---|---|
| EDL | 74.46±0.10 | 76.53±0.14 | 77.40±0.12 | 76.69±0.13 | 71.06±0.10 | 76.28±0.10 | 75.74±0.10 | 76.19±0.10 |
| $\mathcal{I}$-EDL | 82.04±0.10 | 82.48±0.10 | 82.30±0.10 | 82.47±0.10 | 80.63±0.11 | 82.29±0.10 | 81.06±0.09 | 81.96±0.10 |
| R-EDL(Ours) | **83.65±0.10** | **84.22±0.10** | **84.05±0.10** | **84.13±0.10** | **82.39±0.09** | **83.37±0.09** | **82.98±0.09** | **83.28±0.09** |

| Method | 5-Way 20-Shot | | | | 10-Way 20-Shot | | | |
| | MP | UM | DE | MI | MP | UM | DE | MI |
|---|---|---|---|---|---|---|---|---|
| EDL | 80.01±0.10 | 79.78±0.12 | 80.35±0.11 | 79.83±0.12 | 74.50±0.08 | 76.89±0.09 | 76.70±0.08 | 76.86±0.09 |
| $\mathcal{I}$-EDL | 84.29±0.09 | 85.40±0.09 | 85.12±0.09 | 85.35±0.09 | 81.34±0.07 | 82.52±0.07 | 82.16±0.07 | 82.41±0.07 |
| R-EDL(Ours) | **84.85±0.09** | **85.57±0.09** | **85.43±0.09** | **85.53±0.09** | **82.22±0.08** | **82.72±0.07** | **82.48±0.08** | **82.68±0.08** |

Table 10: Comparison of label smoothing and R-EDL in the few-shot setting.

| Method | 5-Way 1-Shot | | 5-Way 5-Shot | | 5-Way 20-Shot | |
| | MP | Ent/UM | MP | Ent/UM | MP | Ent/UM |
|---|---|---|---|---|---|---|
| Label smoothing | 72.03±0.2 | 73.00±0.2 | 77.17±0.2 | 77.11±0.2 | 76.11±0.2 | 75.35±0.2 |
| R-EDL(Ours) | **72.91±0.1** | **74.84±0.1** | **83.65±0.1** | **84.22±0.1** | **84.85±0.1** | **85.57±0.1** |

exclusive; a robust and reliable classifier should excel in both dimensions simultaneously. While both EDL and $\mathcal{I}$-EDL methods tend to excel in only one of the metrics, Table 11 and Fig. 1(a) clearly present the superior performance of R-EDL in terms of the average of these two key metrics.

### D.4 VIDEO-MODALITY SETTING

We also assess our approach using video-modality samples (Bao et al., 2021; Gao et al., 2020), specifically on the open-set action recognition task. Following Bao et al. (2021), we train models on UCF-101 training split and use the testing splits of HMDB-51 and MiT-v2 datasets as unknown sources. Given that the state-of-the-art method DEAR is predicated on EDL, we substitute its EDL implementation with our own R-EDL version. As evidenced by Table 12, this modification yields enhanced performance, further substantiating the efficacy of R-EDL.

Table 11: Results of the noisy setting for VGG16 on CIFAR-10, with the generated noisy data as OOD data, averaged over 5 seeds. Noisy samples are generated by adding zero-mean Gaussian noises with standard deviations of [0.025:0.025:0.200] to the testing samples of CIFAR-10.

| SD of Noise | 0.025 | | | 0.050 | | | 0.075 | | | 0.100 | | |
| Method | Cls | OOD | Avg | Cls | OOD | Avg | Cls | OOD | Avg | Cls | OOD | Avg |
|---|---|---|---|---|---|---|---|---|---|---|---|---|
| EDL | 81.36 (±1.25) | 52.27 (±0.34) | 66.81 | 72.18 (±2.85) | 59.42 (±1.03) | 65.80 | 58.58 (±5.25) | 67.24 (±1.35) | 62.91 | **46.00** (±**5.49**) | 73.08 (±1.81) | 59.54 |
| $\mathcal{I}$-EDL | 85.74 (±0.39) | **54.55** (±**0.58**) | 70.15 | 71.24 (±1.14) | **67.00** (±**1.99**) | 69.12 | 52.14 (±2.65) | **77.33** (±**3.12**) | 64.74 | 36.80 (±3.35) | **83.02** (±**3.71**) | 59.91 |
| R-EDL(Ours) | **87.54** (±**0.35**) | 53.48 (±0.55) | **70.51** | **76.35** (±**1.56**) | 64.19 (±1.02) | **70.27** | **60.19** (±**3.03**) | 75.33 (±1.43) | **67.76** | 45.70 (±3.67) | 82.97 (±1.54) | **64.34** |

| SD of Noise | 0.125 | | | 0.150 | | | 0.175 | | | 0.200 | | |
| Method | Cls | OOD | Avg | Cls | OOD | Avg | Cls | OOD | Avg | Cls | OOD | Avg |
|---|---|---|---|---|---|---|---|---|---|---|---|---|
| EDL | **37.19** (±**4.74**) | 77.28 (±2.24) | 57.23 | **31.55** (±**3.89**) | 80.66 (±2.86) | 56.11 | **27.19** (±**2.77**) | 83.75 (±3.81) | 55.47 | **23.93** (±**2.21**) | 86.58 (±4.67) | 55.26 |
| $\mathcal{I}$-EDL | 26.86 (±3.50) | 85.88 (±4.00) | 56.37 | 20.84 (±3.36) | 87.38 (±4.20) | 54.11 | 17.16 (±3.04) | 88.29 (±4.36) | 52.72 | 15.10 (±2.71) | 88.96 (±4.43) | 52.03 |
| R-EDL(Ours) | 35.16 (±2.97) | **87.62** (±**1.49**) | **61.39** | 28.47 (±1.87) | **90.50** (±**1.55**) | **59.48** | 24.07 (±2.08) | **92.32** (±**1.86**) | **58.20** | 20.83 (±2.50) | **93.50** (±**2.24**) | **57.16** |

Table 12: Results of video-modality setting for I3D backbone on UCF-101, with HMDB-51 and MiT-v2 as OOD data. Results of baselines are reported by Bao et al. (2021).

| Method | UCF-101→HMDB-51 | | UCF-101→MiT-v2 | |
|---|---|---|---|---|
| | Open maF1 | Open Set AUC | Open maF1 | Open Set AUC |
| OpenMax | 67.85±0.12 | 74.34 | 66.22±0.16 | 77.76 |
| MC Dropout | 71.13±0.15 | 75.07 | 68.11±0.20 | 79.14 |
| BNN SVI | 71.57±0.17 | 74.66 | 68.65±0.21 | 79.50 |
| SoftMax | 73.19±0.17 | 75.68 | 68.84±0.23 | 79.94 |
| RPL | 71.48±0.15 | 75.20 | 68.11±0.20 | 79.16 |
| DEAR | 77.24±0.18 | 77.08 | 69.98±0.23 | 81.54 |
| R-EDL(Ours) | **78.73±0.15** | **77.94** | **70.85±0.25** | **82.26** |

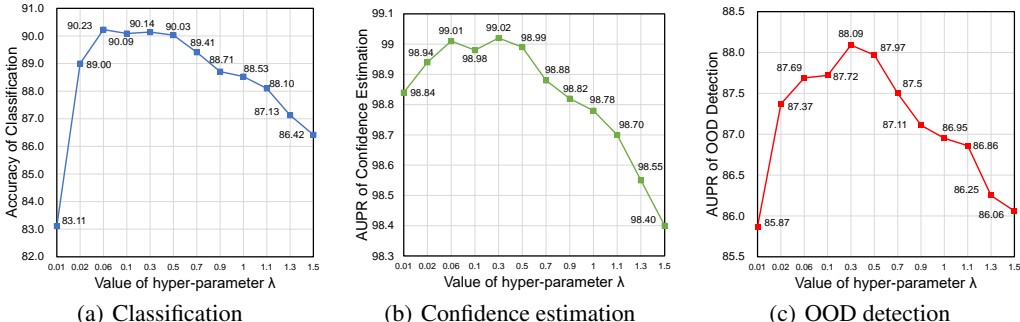

(a) Classification      (b) Confidence estimation      (c) OOD detection

Figure 2: Parameter analysis of the hyper-parameter $\lambda$, (a) evaluated by accuracy of classification on CIFAR-10, (b) AUPR score of confidence estimation on CIFAR-10, and (c) AUPR score of OOD detection against CIFAR-100, respectively.

## D.5 PARAMETER ANALYSIS

Moreover, we further investigate the effect of the hyper-parameter $\lambda$. As an expanded version of figure 1(b), figure 2 includes the trend of variation in classification accuracy on CIFAR-10, the AUPR score for confidence estimation on CIFAR-10, and the AUPR score for OOD detection on CIFAR-100 as the hyper-parameter $\lambda$ varies from 0.01 to 1.5. Observation reveals that a smaller $\lambda$ generally outperforms the tradition EDL setting where $\lambda = 1$, indicating that the EDL setting for the prior weight is excessively high, leading to suboptimal results. Nonetheless, an excessively small $\lambda$ also has detrimental effects. For instance, setting $\lambda$ to 0.01 results in a significant decrease in classification accuracy to 83.11% and a drop in the AUPR score of OOD detection to 85.87%. In this setting, $\lambda$ is ultimately established at 0.1, selected from the range [0.1:0.1:1.0] based on the best classification accuracy on the validation set.

## D.6 VISUALIZATION OF UNCERTAINTY DISTRIBUTIONS

Figs. 3,4,5, and 6 show density plots of the normalized uncertainty measures for CIFAR-10 against SVHN, and CIFAR-10 against CIFAR-100, while Figs. 7,8,9, and 10 show density plots for MNIST against FMNIST, and MNIST against KMNIST. The uncertainty measures include max projected probability, uncertainty mass, differential entropy, and mutual information. We apply min-max normalization on each uncertainty value $u$, i.e., $u_{norm} = (u - \min u)(\max u - \min u)$. It can be observed that our method attaches higher confidence to ID data and makes uncertainty of OOD data more aggregated, exhilarating better separability.

The density plots of I-EDL show different shapes with those of EDL and R-EDL, since I-EDL utilizes the Fisher information matrix to measure the amount of information that the categorical probabilities carry about the concentration parameters of the corresponding Dirichlet distribution, thus allowing a certain class label with higher evidence to have a larger variance. Consequently, the predictions made by the I-EDL approach are typically less extreme, resulting in a bimodal distribu-

tion on the uncertainty density plot where the two peaks are closer to the center of the density axis compared to the EDL and R-EDL methods.

Additionally, we deduce that the similarity in the shapes of the uncertainty density plots between EDL and R-EDL may stem from the fact that R-EDL's modifications to EDL only consist of relaxations of non-essential settings, without introducing any additional mechanisms.

## E  SOURCE CODE

Our code is available at https://github.com/MengyuanChen21/ICLR2024-REDL.

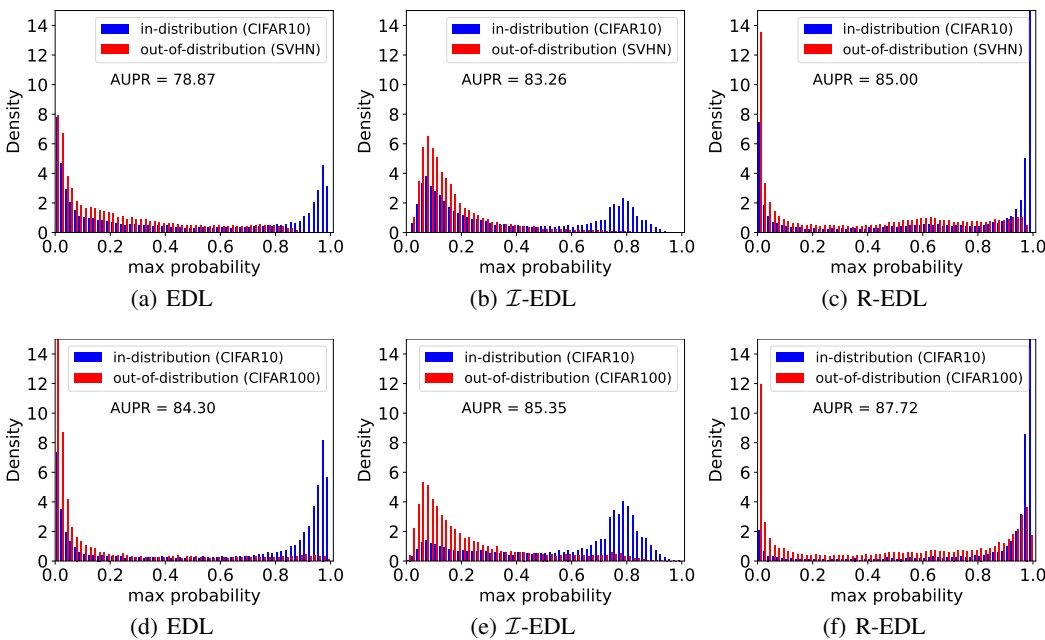

Figure 3: Uncertainty distribution measured by max projected probability on CIFAR10.

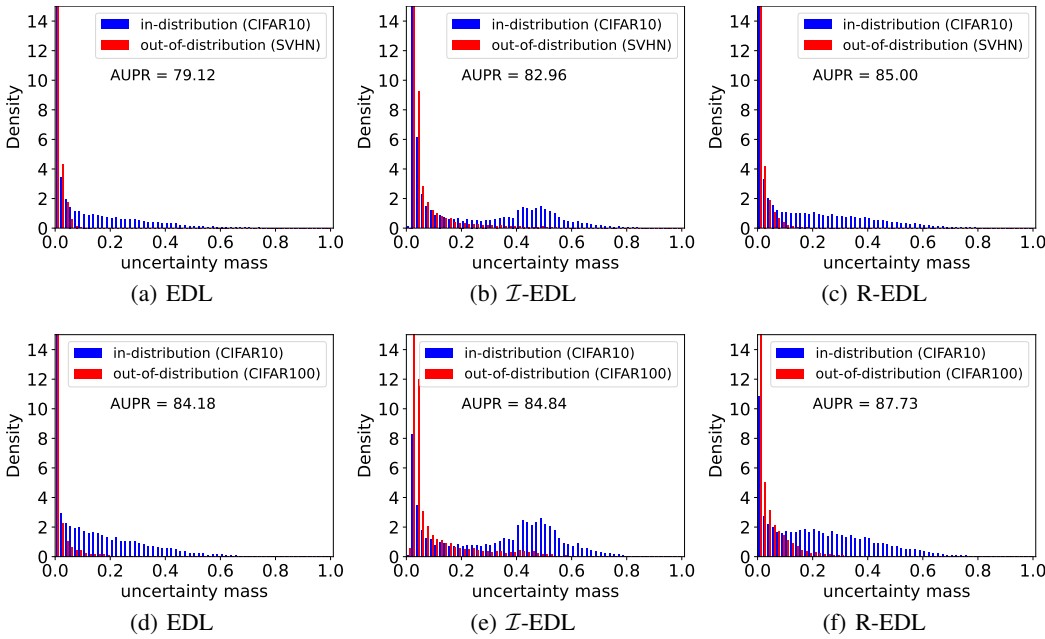

Figure 4: Uncertainty distribution measured by uncertainty mass on CIFAR10.

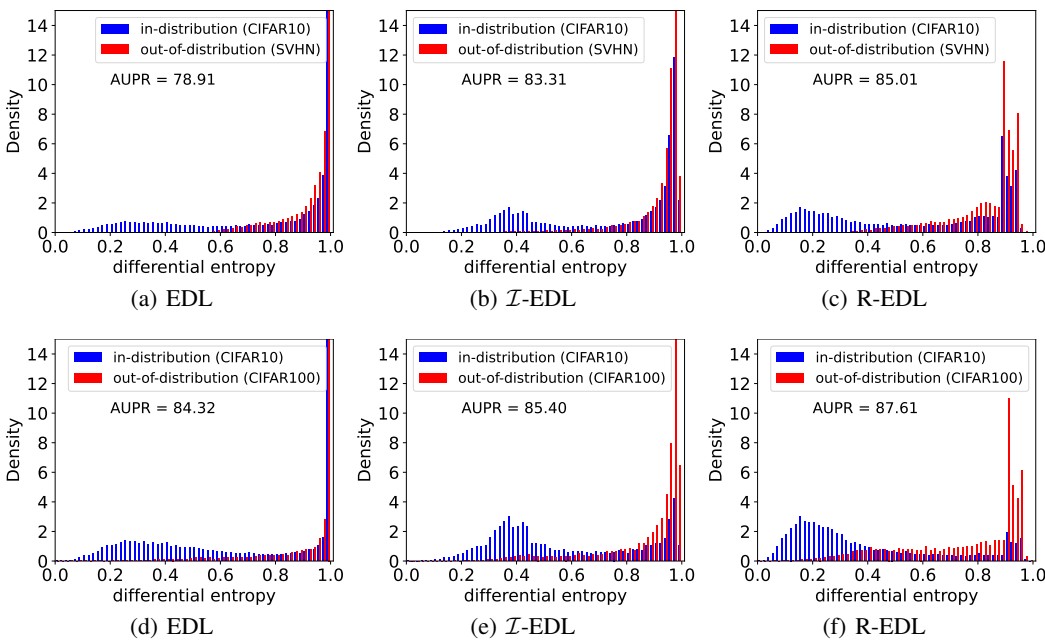

Figure 5: Uncertainty distribution measured by differential entropy on CIFAR10.

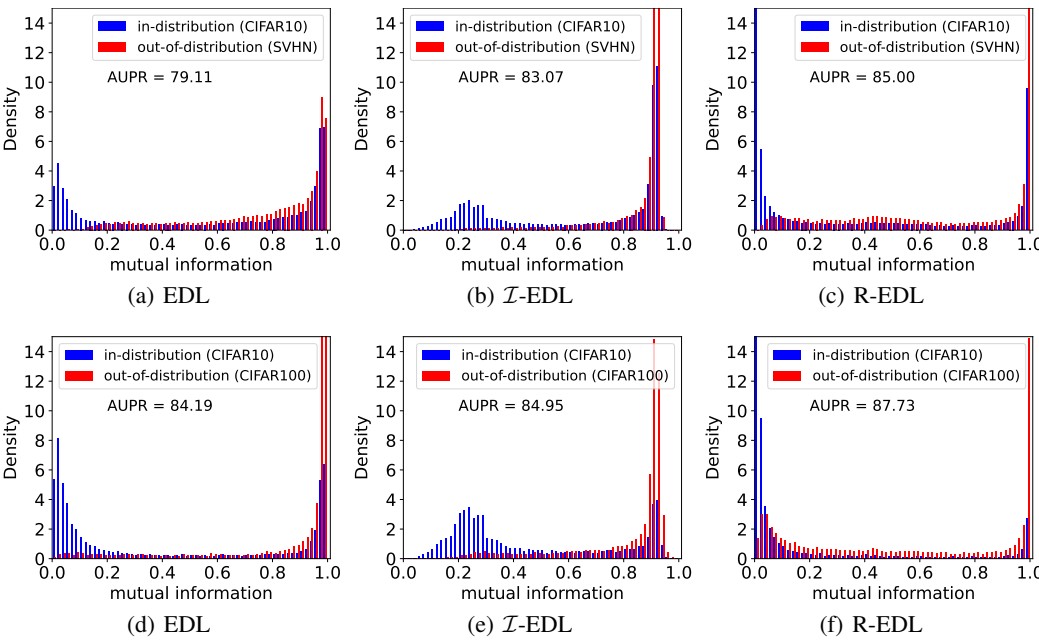

Figure 6: Uncertainty distribution measured by mutual information on CIFAR10.

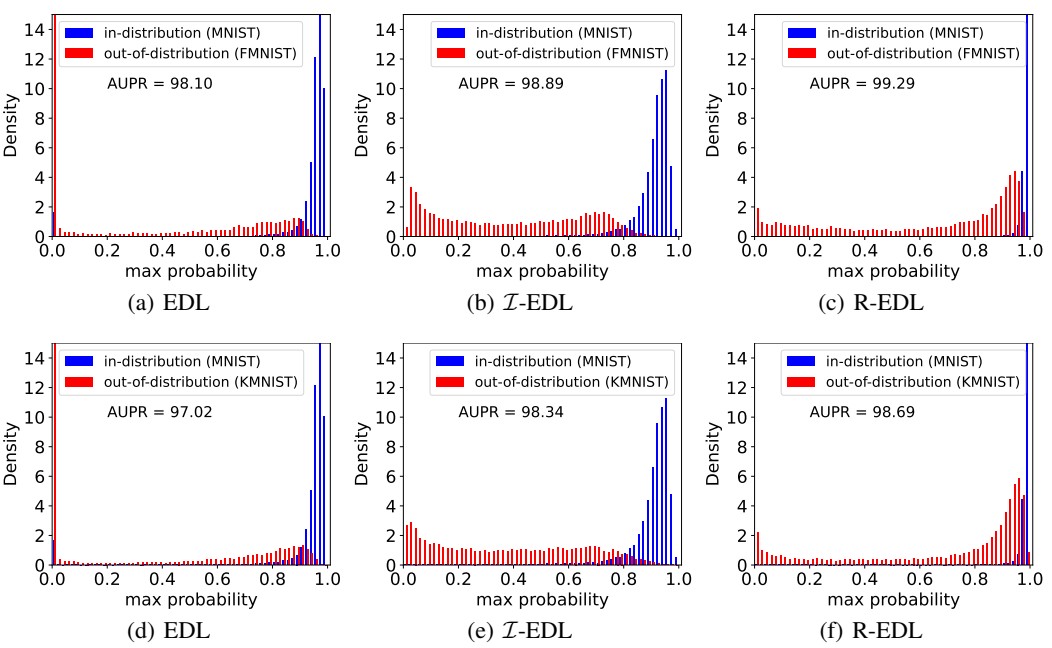

Figure 7: Uncertainty distribution measured by max projected probability on MNIST.

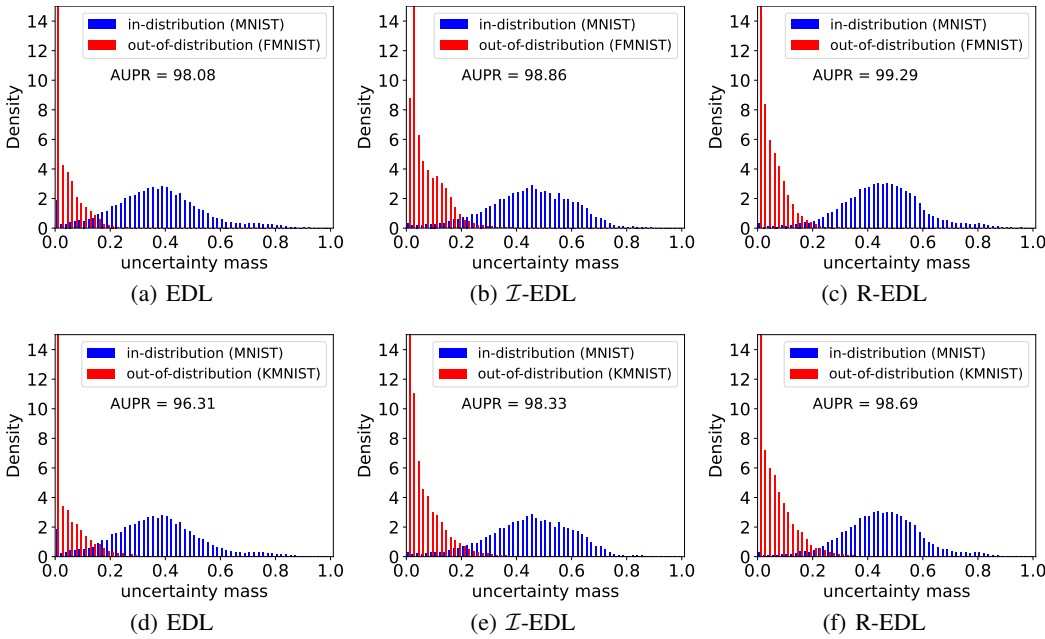

Figure 8: Uncertainty distribution measured by uncertainty mass on MNIST.

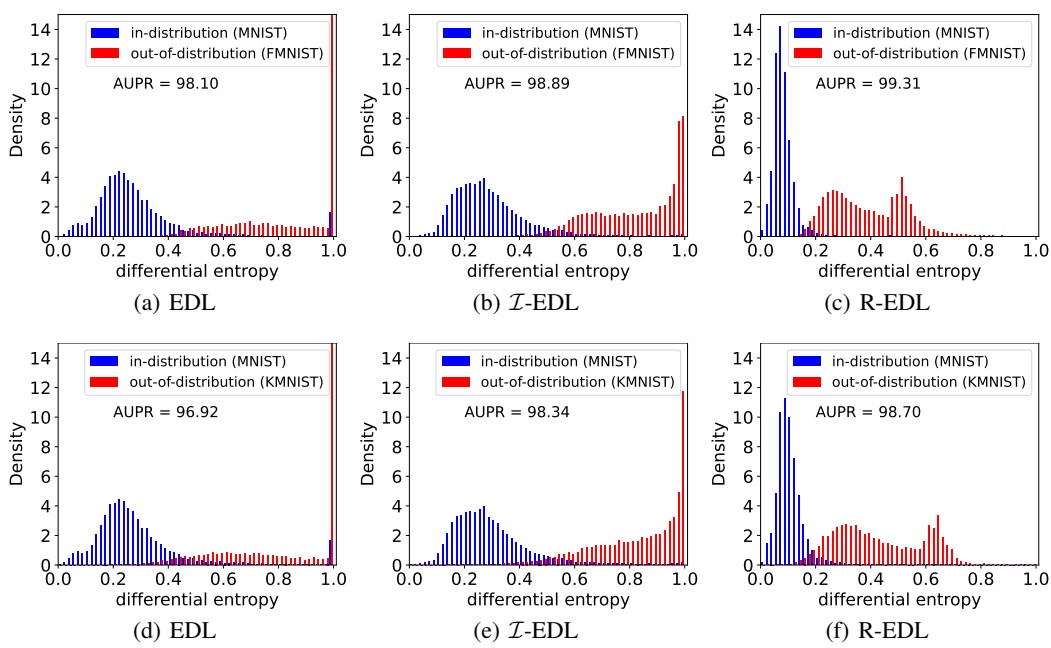

Figure 9: Uncertainty distribution measured by differential entropy on MNIST.

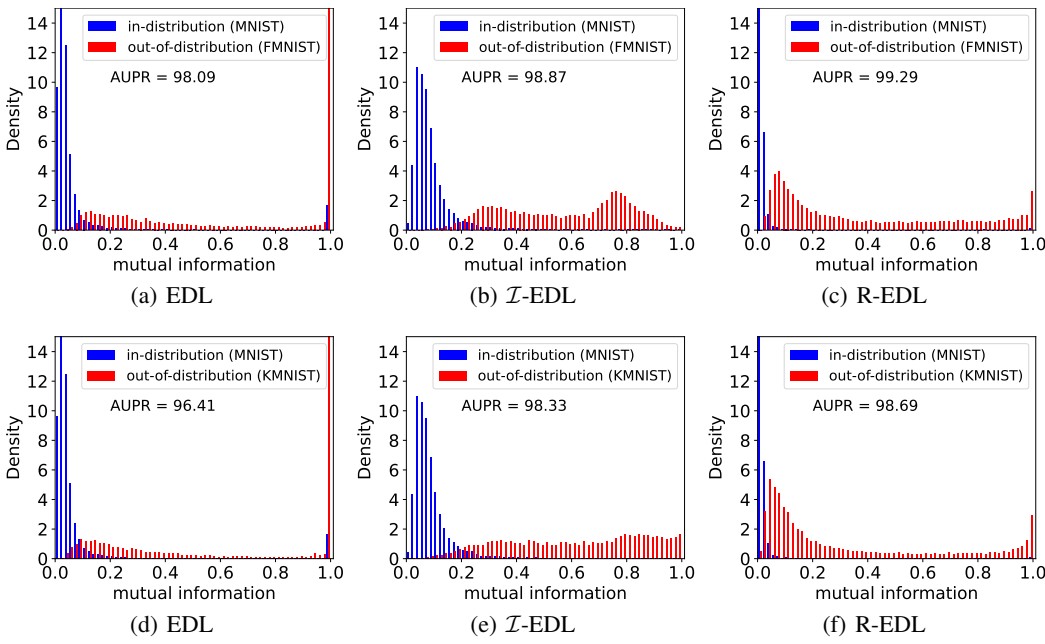

Figure 10: Uncertainty distribution measured by mutual information on MNIST.

