# OpenReview forum: "R-EDL: Relaxing Nonessential Settings of Evidential Deep Learning"
_ICLR.cc/2024/Conference — ICLR 2024 spotlight_

### Official Review · Reviewer_R4i1 · 2023-10-29

**Soundness:** 3 good
**Presentation:** 3 good
**Contribution:** 3 good
**Rating:** 6
**Confidence:** 3

**Summary:**

The authors propose a novel R-EDL method to recalibrate uncertainty estimation by relaxing nonessential settings of EDL.

**Strengths:**

- An interesting perspective to explore uncertainty estimation

- Comprehensive experiments

**Weaknesses:**

- Could you present the detailed derivation of Formulas 7,9 and 10?  These formulas are the essential steps, but I may not catch up with your idea.

- The authors are recommended to make necessary explanations on different shapes of I-EDL with EDL and R-EDL in Figure 2 and Figure 5.
Additionally, uncertainty representations for ID (MNIST) and OOD (FMNIST) are better to show the differences among EDL, I-EDL, and R-EDL, as presented in I-EDL (figure 4).

- What role did W play in the process of uncertainty estimation or evidence collection?

**Questions:**

as aforementioned

---

> ### Author Response · Authors · 2023-11-18
> **Response to Reviewer R4i1**
>
> Thank you for your constructive comments. We have meticulously revised the manuscript as suggested.  In the following, your comments are first stated and then followed by our point-by-point responses.
>
> > Comment 1: Could you present the detailed derivation of Formulas 7,9 and 10? These formulas are the essential steps, but I may not catch up with your idea.
>
> Thanks for the comment. Presented below are the step-by-step derivations of Formulas 7, 9, and 10, respectively. For the sake of brevity, some explanation of notations, which have been provided in the main text, have been omitted here.
>
> Formula 7 is a generalized form of the projected probability $P_X$, where the prior weight $W$ is not fixed to the class number $C$. Its derivation is presented below:
> - In Definition 2, the projected probability $P_X$ of the opinion $\omega_X$ is defined by $P_X(x) = b_X(x) + a_X(x) u_X$.
> - Without fixing the prior weight $W$ to $C$, Formula 5 will be transformed into  $b_X(x) = \frac{e_X(x)}{\sum_{x^\prime\in \mathbb X}e_X(x^\prime) + W},\quad u_X= \frac{W}{\sum_{x\in \mathbb X}e_X(x) + W}$.
> - By setting the base rate $a_X(x)$ as a uniform distribution, i.e., $a_X(x)=1/C$ and plugging the above $b_X(x)$ and $u_X$ into the definition of $P_X$, we come to formula 7: $P_X(x)=\frac{e_X(x) + \frac{W}{C}}{\sum_{x^\prime\in \mathbb X}e_X(x^\prime) + W}$.
>
> Formula 9 is a generalized form of the concentration parameter $\alpha_X$ of the constructed Dirichlet PDF, whose derivation is presented below:
> - Formula 4 shows $\alpha_X(x) = \frac{b_X(x) W}{u_X} + a_X(x)W$.
> - By setting the base rate $a_X(x)$ as a uniform distribution, i.e., $a_X(x)=1/|\mathbb{X}|=1/C$, and plugging the above $b_X(x)$ and $u_X$ into Formula 4, we have $\alpha_X(x) = e_X(x) + \frac{W}{C}=e_X(x)+\lambda$, where $\lambda=W/C$.
>
> Formula 10 is equivalent to Formula 7. We reiterated it in Section 3.3 to facilitate subsequent discussion about the optimization objective. Employing Formula 9, its derivation can be more easily achieved:
>
> - Formula 9 shows $\alpha_X(x)=e_X(x)+\lambda$.
> - As defined below Formula 6，$S_X$ denotes the sum of $\alpha_X(x)$, thus we have $S_X = \sum_{x^\prime\in \mathbb X}e_X(x^\prime) + C\lambda$.
> - Then we come to Formula 10: $P_X(x) = \frac{\alpha_X(x)}{S_X}=\frac{e_X(x) + \lambda}{\sum_{x^\prime\in \mathbb X}e_X(x^\prime) + C\lambda}$.
>
> We hope that the above derivation process will aid in the comprehension of our work. Should any aspects of our paper remain ambiguous, we warmly welcome further discussion.
>
> > Comment 2: The authors are recommended to make necessary explanations on different shapes of I-EDL with EDL and R-EDL in Figure 2 and Figure 5. Additionally, uncertainty representations for ID (MNIST) and OOD (FMNIST) are better to show the differences among EDL, I-EDL, and R-EDL, as presented in I-EDL (figure 4).
>
> Thanks for the constructive comment! **The necessary explanations and the uncertainty density plots (Figure 7-10) for ID (MNIST) and OOD (FMNIST) have been added to the appendix of the revised version.**
>
> The explanations on different shapes of I-EDL with EDL and R-EDL are as follows: The I-EDL method utilizes the Fisher information matrix to measure the amount of information that the categorical probabilities carry about the concentration parameters of the corresponding Dirichlet distribution, thus allowing a certain class label with higher evidence to have a larger variance. Consequently, the predictions made by the I-EDL approach are typically less extreme, resulting in a bimodal distribution on the uncertainty density plot where the two peaks are closer to the center of the density axis compared to the EDL and R-EDL methods. Additionally, we deduce that the similarity in the shapes of the uncertainty density plots between EDL and R-EDL may stem from the fact that R-EDL's modifications to EDL only consist of relaxations of non-essential settings, without introducing any additional mechanisms.
>
> > Comment 3: What role did W play in the process of uncertainty estimation or evidence collection?
>
> Thanks for the comment.
>
> Theoretically, the existence of $W$ serves as a necessary condition for the bijective relationship between subjective opinions and Dirichlet PDFs in Theorem 1. As shown in Formula 4, a single subjective opinion could correspond to multiple Dirichlet PDFs in the absence of $W$.
>
> Empirically, as elaborated in Section 3.2, the value of $W$ determines the extent to which the projected probability $P_X(x)$ is influenced by the magnitude and proportion of evidence respectively. As one of our key contributions, relaxing the rigid setting of fixing $W$ to the number of classes has been shown to enhance the quality of uncertainty estimation.
>
> **In the end, thanks for all your time and consideration in reviewing our paper!**

---

> ### Author Response · Authors · 2023-11-22
> **Looking forward to your reply**
>
> Dear Reviewer R4i1,
>
> We have provided new results and discussions according to your valuable suggestions, which have been absorbed into the revised manuscript. We believe that the revised manuscript has been strengthened by incorporating your suggestions.
>
> As the rebuttal deadline is approaching, we look forward to your reply or any other suggestions. Thank you so much!
>
> Best Regards,
>
> Authors of Submission 597

---

> > ### Comment · Reviewer_R4i1 · 2023-11-23
> > **Sincerely thanks for your support and raising the score**
> >
> > Thanks for your detailed reply. My concerns are addressed and I have raised my score.
> >
> > Best

---

### Official Review · Reviewer_5NaH · 2023-10-30

**Soundness:** 3 good
**Presentation:** 2 fair
**Contribution:** 3 good
**Rating:** 6
**Confidence:** 3

**Summary:**

The paper proposes new tweaks to the existing framework of evidential deep learning. These tweaks include treating a pre-set parameter as a tuning hyperparameter. The paper also uses an alternative objective to optimize for evidential deep learning.

The paper provides mathematical justification for the design choices. The paper also run experiments to demonstrate that the tweaks are effective.

**Strengths:**

originality: the paper provides incremental improvement toward the existing evidential deep learning framework.
quality: the proposed modification seems solid both mathematically and empirically. I particularly like that the paper carried out fairly exhaustive experiments to measure the utility of the proposed method across various tasks, comparing the proposed method with various alternatives. The ablation study is also a nice addition.
clarity: the paper provides a mathematical formula and associated explanation to justify the tweaks of the framework presented in the paper. I can follow the arguments made in the paper on the high level.
significance: the proposed tweaks appeared to make the framework more pratical and effective.

**Weaknesses:**

* presentation can be improved.
  * for example, more intuition about some of the key concepts of the paper can be provided such as subjective opinion, belief mass, uncertainty mass, and Dirichlet distribution.
  * In the experiments, max probability and UM are used as confidence scores. However, it is not clear how to interpret them and what is a good score and a bad one.
  * Table 1 caption mentioned AUPR score, but it does not seem to be actually reported.
* Consider the prior weight as hyperparameter also means that the algorithm requires hyperparameter tunning.

**Questions:**

* "Despite its great success" sentence in the abstract is difficult to parse. Consider breaking it into a few sentences.

---

> ### Author Response · Authors · 2023-11-18
> **Response to Reviewer 5NaH**
>
> We express our gratitude for your comments. We have meticulously revised the manuscript as suggested.  Below, we first reiterate your comments, subsequently providing our detailed responses to each point.
>
> > Comment 1: Presentation can be improved. For example, more intuition about some of the key concepts of the paper can be provided such as subjective opinion, belief mass, uncertainty mass, and Dirichlet distribution.
>
> Thanks for the valuable comment. Your comment serves as a crucial reminder that some concepts in the paper may be perplexing to readers. Here we provide some useful intuition about these concepts:
>
> - **Subjective opinions** are based on **belief mass** distributions over a domain $\mathbb{X}$. **Belief mass** assigned to a singleton value $x\in\mathbb{X}$ expresses support for $x$ being TRUE.
> - **Belief mass** distributions are subadditive, meaning that the sum of **belief massess** can be less than one. The sub-additivity of the **belief mass** distribution is complemented by **uncertainty mass**.
> - The **uncertainty mass** represents vacuity of evidence, i.e., the lack of support for the variable $X$ to have any specific value. It can also be interpreted as **belief mass** assigned to the entire domain.
> - The sub-additivity of the **belief mass** distribution and the complement property of the **uncertainty mass** are expressed by the additivity requirement in Definition 1.
> - An important aspect of **subjective opinions** is that they are equivalent to Beta or **Dirichlet PDFs** (probability density functions) under a specific mapping, which has been elaborated on by Theorem 1.
>
> **A more explicit explanation of belief mass and uncertainty mass has been added to the revised version (below Definition 1).** Besides, **we have also incorporated necessary guidance in the appendix**, directing readers to references [1,2] for a more comprehensive understanding. Thanks again for the reminder!
>
> [1] Jøsang, Audun. A logic for uncertain probabilities. International Journal of Uncertainty, Fuzziness and Knowledge-Based Systems, 2001.
>
> [2] Jøsang, Audun. Subjective logic. Vol. 3. Cham: Springer, 2016.
>
> > Comment 2: In the experiments, max probability and UM are used as confidence scores. However, it is not clear how to interpret them and what is a good score and a bad one. Table 1 caption mentioned AUPR score, but it does not seem to be actually reported.
>
> Thank you for the comment, but there appears to be some misunderstanding.
>
> **The scores listed beneath "MP" and "UM" are all AUPR scores**. MP and UM are two choices of confidence scores, and then they are used to compute the AUPR, which evaluates the model's ability of discriminating classified and misclassified samples, as well as ID and OOD samples.
>
> Besides, regarding the question which is the better option for confidence scores between "MP" and "UM", as indicated by Table 1, there is no absolute consensus. Instead, the preferable choice depends on the specific methods, datasets, and tasks involved.
>
> > Comment 3: Consider the prior weight as hyperparameter also means that the algorithm requires hyperparameter tunning.
>
> Thanks for the insightful comment. Indeed, our algorithm necessitates the tuning of hyperparameters. Nonetheless, as Figure 1(b) demonstrably illustrates, the algorithm consistently exhibits satisfactory performance, not merely at an isolated hyperparameter value, but across a continuous range. This indicates that the process of hyperparameter tuning should not pose significant challenges.
>
> > Comment 4: "Despite its great success" sentence in the abstract is difficult to parse. Consider breaking it into a few sentences.
>
> Thanks for the suggestion. We guess what you refer to is the "In this work, our analysis" sentence in the abstract. We have modified its expression as suggested in the revised version.
>
> **In the end, thanks for all your time and consideration!**

---

> ### Author Response · Authors · 2023-11-22
> **Looking forward to your reply**
>
> Dear Reviewer 5NaH,
>
> We have provided new discussions according to your valuable suggestions, which have been absorbed into the revised manuscript. We believe that the revised manuscript has been strengthened by incorporating your suggestions.
>
> As the rebuttal deadline is approaching, we look forward to your reply or any other suggestions. Thank you so much!
>
> Best Regards,
>
> Authors of Submission 597

---

### Official Review · Reviewer_vQHP · 2023-10-30

**Soundness:** 3 good
**Presentation:** 3 good
**Contribution:** 3 good
**Rating:** 8
**Confidence:** 5

**Summary:**

The paper improves the Evidential Deep Learning (EDL) approach, a method for performing classification with reliable uncertainty estimations, in two ways: i) adopting a more generic definition of evidence that also incorporates prior belief, ii) removing the Kullback-Leibler (KL) divergence based regularizer term and employing a prior weight term for regularization that emerges naturally from this generalized definition.

**Strengths:**

* The proposed approach for improving EDL is novel, creative, and innovative. It is also built on solid theoretical foundations. The resulting loss function is derived rigorously from first principles and clearly identifies the limitations of EDL.

* This is a high-quality piece of work that not only develops a novel method, but also illustrates its development steps and the rationale of each step unequivocally, as well as demonstrating the practical benefit of the end outcome with a large body of experiments. The proposed method brings a clear performance improvement.

**Weaknesses:**

*  The paper below addresses the over/under regularization problem of EDL using an alternative approach, strictly speaking via Bayesian inference: **“Kandemir et al., Evidential Turing Processes, ICLR, 2022”**. It could be worthwhile to clarify the advantage of the proposed approach with respect to this alternative.

* The phrase “multiple mutually disjoint values” in Definition 1 is a little confusion and potentially imprecise. A better alternative could be **“Let X be a categorical random variable on the domain $\mathbb{X}$”**

**Questions:**

* While the paper justifies all the steps up till Equation 10 rigorously, the choice of the loss in Equation 11 looks a bit arbitrary. Why should one choose the MSE between the class probabilities and one-hot labels instead of the simple and straightforward cross-entropy loss? How would it work numerically? If it works better or worse, what is the interpretation of the outcome?

* Why were the MP and UM scores chosen as the performance metrics to measure in-distribution performance instead of the well-established Expected Calibration Error (ECE)?

* How does the method compare against post-hoc calibration algorithms such as temperature scaling?

---

> ### Author Response · Authors · 2023-11-18
> **Response to Reviewer vQHP (Part 1/3)**
>
> We express our sincere gratitude for your insightful and constructive comments. We have meticulously revised the manuscript as suggested.  Below, we first reiterate your comments, subsequently providing our detailed responses to each point.
>
> > Comment 1: The paper below addresses the over/under regularization problem of EDL using an alternative approach, strictly speaking via Bayesian inference: **“Kandemir et al., Evidential Turing Processes, ICLR, 2022”**. It could be worthwhile to clarify the advantage of the proposed approach with respect to this alternative.
>
> We extend our gratitude for directing our attention to this important work.  A brief discussion of the advantage and the disadvantage of this alternative has been added to the Related Work section in the revised version as follows:
>
> "Kandemir et al. (2022) combines EDL, neural processes, and neural Turing machines to propose the Evidential Tuning Process, which shows stronger performances than EDL but requires a rather complex memory mechanism."
>
> > Comment 2: While the paper justifies all the steps up till Equation 10 rigorously, the choice of the loss in Equation 11 looks a bit arbitrary. Why should one choose the MSE between the class probabilities and one-hot labels instead of the simple and straightforward cross-entropy loss? How would it work numerically? If it works better or worse, what is the interpretation of the outcome?
>
> Thanks for the insightful comment!
>
> When choosing which loss function to optimize the projected probability, we followed the previous EDL-related works [1],[2] to use the Mean Squared Error (MSE) loss without extensive additional exploration. Empirically, we find that MSE results in better performance than Cross-Entropy (CE) loss, which is also mentioned by [1], particularly when employing a non-exponential evidence activation function. In our study, as detailed in the appendix, , we use Softplus as the evidence activation function.
>
> In contrast, when $\exp$ is used as the evidence activation function, as seen in EDL-related works [3] and [4], the CE loss is commonly preferred over the MSE loss. However, we noted that the use of the exponential function as the evidence activation leads to significant fluctuations in the value of the evidence, complicating the tuning of our introduced parameter. Therefore, based on these observations and considerations, we selected the Softplus function as our evidence activation function and the MSE as the loss function form for our model.
>
> [1] Sensoy, Murat, Lance Kaplan, and Melih Kandemir. Evidential deep learning to quantify classification uncertainty. NeurIPS, 2018.
>
> [2] Deng, Danruo, et al. Uncertainty estimation by fisher information-based evidential deep learning. ICML, 2023.
>
> [3] Bao, Wentao, Qi Yu, and Yu Kong. Evidential deep learning for open set action recognition. ICCV, 2021.
>
> [4] Chen, Mengyuan, et al. Dual-evidential learning for weakly-supervised temporal action localization. ECCV, 2022.

---

> ### Author Response · Authors · 2023-11-18
> **Response to Reviewer vQHP (Part 2/3)**
>
> > Comment 3: Why were the MP and UM scores chosen as the performance metrics to measure in-distribution performance instead of the well-established Expected Calibration Error (ECE)?
>
> Thanks for the insightful comment!
>
> Strictly speaking, we are using the area under the precision-recall curve (AUPR) of discriminating correctly classified and misclassified samples to measure in-distribution confidence estimation performance. MP and UM are two choices of the confidence scores for calculating the AUPR score. To elucidate our rationale for adopting the AUPR as the evaluation metric, as suggested in reference **"Deng, Danruo, et al. Uncertainty estimation by fisher information-based evidential deep learning. ICML, 2023"**, over the Brier Score or ECE, we present our justification below.
>
> As delineated in Section 5.2, our stance is that a reliable uncertainty estimator should **(1)** assign higher uncertainties to misclassified samples than to correctly classified ones, and **(2)** assign higher uncertainties to out-of-distribution (OOD) samples than to in-distribution (ID) ones. In other words, **our primary criterion for assessing confidence estimation is the model's ability in differentiating between correctly classified and misclassified samples, as well as between ID and OOD samples based on the predicted confidence**. This is the fundamental reason for our selection of AUPR as the evaluation metric. The term "over-confidence," as used in our context, specifically pertains to the tendency of neural networks to assign excessively high confidence to misclassified or OOD samples, a perspective that deviates from some existing literature.
>
> We recognize that ECE is frequently employed to assess the degree of correspondence between the model's confidence and the true correctness likelihood, which is important for classification models in many applications. However, **a confidence distribution accompanied by a low ECE does not inherently ensure a clear distinction between correct and incorrect predictions**. For instance, in a balanced two-class dataset scenario, if a binary classifier categorizes all samples into a single class with a consistent confidence output of 50%, the ECE would be zero, yet this result lacks practical significance. Therefore, we posit that for the objectives of our study, AUPR provides a more pertinent evaluation. Furthermore, whereas ECE is focused on the absolute values of confidence, AUPR concentrates on the ranking order of confidence. We consider both metrics as direct evaluators of the quality of predicted confidence, each with its unique perspective and utility.
>
> | Method | ECE (15 bins)$\downarrow$ | Brier score$\downarrow$ | AUPR (confidence) | AUPR (OOD against SVHN)
> | :---: |:---: |:---: |:---:|:---:|
> | EDL | 20.08$\pm$1.77 | 40.68$\pm$2.39 |97.86$\pm$0.17|78.87$\pm$3.50|
> |I-EDL| 39.96$\pm$0.37 | 55.32$\pm$0.50 |98.72$\pm$0.12|83.26$\pm$2.44|
> |R-EDL| **3.48**$\pm$0.30 | **18.14**$\pm$0.51 |**98.98**$\pm$0.05|**85.00**$\pm$1.22|
>
> As shown in the above table, which summarizes results from five runs on CIFAR10, our proposed method outperforms EDL and I-EDL with a large margin when evaluated by the ECE and Brier score. It is mainly due to that we relax the rigid setting of fixing the prior weight parameter to the number of classes. With the original setting of $W$ (or $\lambda$), as illustrated in Section 3.2, it is difficult for previous EDL-related works to make predictions of high confidence, despite the fact that they actually output correct evidence distributions, thus resulting in excessively high ECE and Brier scores. With our modification, model's predicted confidence becomes much more aligned with the expected classification accuracy. It is noteworthy that, although previous EDL-related works paid little attention to the correspondence of confidence and expected accuracy, they are still effective in detecting misclassified and OOD samples, which also aligns with our primary objective.
>
> In summary, our stance is that **employing the AUPR scores for evaluation purposes aligns more closely with our objectives than using ECE or Brier score**. Additionally, for the sake of comprehensive analysis and transparency, **the results evaluated using ECE and Brier score have been included in the appendix of the revised version**, serving as a reference point.

---

> ### Author Response · Authors · 2023-11-18
> **Response to Reviewer vQHP (Part 3/3)**
>
> > Comment 4: How does the method compare against post-hoc calibration algorithms such as temperature scaling?
>
> | Method | ECE (15 bins) $\downarrow$ | Brier Score $\downarrow$ | AUPR (confidence) | AUPR (OOD against SVHN) |
> | :---: |:---: |:---: |:---:|:---:|
> |Temp Scale | **1.06**$\pm$0.10 | 18.44$\pm$0.49 | 98.89$\pm$0.05 | 81.89$\pm$2.19 |
> | EDL | 20.08$\pm$1.77 | 40.68$\pm$2.39 | 97.86$\pm$0.17 | 78.87$\pm$3.50 |
> |I-EDL| 39.96$\pm$0.37 | 55.32$\pm$0.50 | 98.72$\pm$0.12 | 83.26$\pm$2.44 |
> |R-EDL| 3.48$\pm$0.30  | **18.14**$\pm$0.51 | **98.98**$\pm$0.05 | **85.00**$\pm$1.22 |
>
> We appreciate the constructive comment, and **the comparison between our method and the temperature scaling method has been added to the appendix of our revised version**.
>
> The results of temperature scaling, derived from five runs on the CIFAR10 dataset, is reported above. We utilized the [official code](https://github.com/gpleiss/temperature_scaling) of temperature scaling to implement this method, which indeed exhibits excellent performances when measured by ECE. However, there still exists a performance gap in OOD detection when compared to our method. Besides, as stated above, employing the AUPR scores for evaluation aligns more closely with our objectives than using ECE.
>
> > Comment 5: The phrase “multiple mutually disjoint values” in Definition 1 is a little confusion and potentially imprecise. A better alternative could be **“Let $X$ be a categorical random variable on the domain $\mathbb{X}$**”.
>
> Thanks for the valuable comment. We have modified the expression as suggested in the revised version.
>
> **In the end, we express our gratitude for your time and consideration. We are also immensely thankful for your constructive suggestions and the positive rating.**

---

> > ### Comment · Reviewer_vQHP · 2023-11-23
> > **Thanks**
> >
> > Thanks for answering my questions with great rigor. All my open points are addressed now. I keep my score as an accept.

---

### Official Review · Reviewer_yPVK · 2023-11-02

**Soundness:** 3 good
**Presentation:** 3 good
**Contribution:** 3 good
**Rating:** 8
**Confidence:** 4

**Summary:**

The authors focus on improving the popular Evidential Deep Learning approach introduced by Sensoy et al. (2018). They identify two potential sources of improving it, (i) the prior weight by turning it into a hyperparameter, and (ii) the variance term in the objective loss, by removing it. The approach is evaluated in a series of experiments.

**Strengths:**

- The paper is well written with a clear line of thought and additional implementations.
- Both modifications, prior weight, and removal of variance regularization, are motivated theoretically and their respective influence is validated via an ablation study.
- The improvements are evaluated in a series of experiments

**Weaknesses:**

- The experiments lack an in-distribution calibration evaluation



### Minor
- Typo: The first paragraph on p7 KL-PN is mentioned twice

**Questions:**

- Can the authors report additional calibration results? E.g., by reporting the expected calibration error (Guo et al., 2017)?

---

> ### Author Response · Authors · 2023-11-18
> **Response to Reviewer yPVK**
>
> Thanks for your constructive comments and positive rating. We have meticulously revised the manuscript as suggested.  In the following, your comments are first stated and then followed by our point-by-point responses.
>
> > Comment 1: The experiments lack an in-distribution calibration evaluation. Can the authors report additional calibration results? E.g., by reporting the expected calibration error (Guo et al., 2017)?
>
> Thanks for the constructive comment! **The additional results have now been reported in the appendix of the revised version.**
>
> Instead of performing an in-distribution calibration evaluation, we have selected AUPR scores to evaluate whether the model can assign higher confidence to correctly classified samples than misclassified ones. We would like to provide more explanation for our choice when reporting the results evaluated by ECE and Brier score.
>
> | Method | ECE (15 bins)$\downarrow$ | Brier score$\downarrow$ | AUPR (confidence) | AUPR (OOD against SVHN)
> | :---: |:---: |:---: |:---:|:---:|
> | EDL | 20.08$\pm$1.77 | 40.68$\pm$2.39 |97.86$\pm$0.17|78.87$\pm$3.50|
> |I-EDL| 39.96$\pm$0.37 | 55.32$\pm$0.50 |98.72$\pm$0.12|83.26$\pm$2.44|
> |R-EDL| **3.48**$\pm$0.30 | **18.14**$\pm$0.51 |**98.98**$\pm$0.05|**85.00**$\pm$1.22|
>
> As shown in the above table, which summarizes results from five runs on CIFAR10, our proposed method outperforms EDL and I-EDL with a large margin when evaluated by the ECE and Brier score. It is mainly due to that we relax the rigid setting of fixing the prior weight parameter to the number of classes. With the original setting of $W$ (or $\lambda$), as illustrated in Section 3.2, it is difficult for previous EDL-related works to make predictions of high confidence, despite the fact that they actually output correct evidence distributions, thus resulting in excessively high ECE and Brier scores. With our modification, model's predicted confidence becomes much more aligned with the expected classification accuracy. It is noteworthy that, although previous EDL-related works paid little attention to the correspondence of confidence and expected accuracy, they are still effective in detecting misclassified and OOD samples, which also aligns with our primary goal.
>
> As delineated in Section 5.2, our stance is that a reliable uncertainty estimator should (1) assign higher uncertainties to misclassified samples than to correctly classified ones, and (2) assign higher uncertainties to OOD samples than to ID ones. In other words, our primary criterion for assessing confidence estimation is the model's ability in differentiating between correctly classified and misclassified samples, as well as between ID and OOD samples based on the predicted confidence. This is the fundamental reason for our selection of AUPR as the evaluation metric. The term "over-confidence," as used in our context, specifically pertains to the tendency of neural networks to assign excessively high confidence to misclassified or OOD samples, a perspective that deviates from some existing literature.
>
> We recognize that ECE is frequently employed to assess the degree of correspondence between the model's confidence and the true correctness likelihood, which is important for classification models in many applications. However, a confidence distribution accompanied by a low ECE does not inherently ensure a clear distinction between correct and incorrect predictions.
>
> > Comment 2: Typo: The first paragraph on p7 KL-PN is mentioned twice.
>
> Thanks for the reminder! The second "KL-PN" should be "RKL-PN". It has been corrected.
>
> **In the end, we extend our sincerest thanks for all your time and consideration! Thanks again for your constructive suggestions and postive rating!**

---

> > ### Comment · Reviewer_yPVK · 2023-11-21
> >
> > Thank you for your detailed answers to the other reviewers an me. I keep my score of recommending acceptance.

---

### Official Review · Reviewer_ErzN · 2023-11-07

**Soundness:** 3 good
**Presentation:** 4 excellent
**Contribution:** 3 good
**Rating:** 8
**Confidence:** 4

**Summary:**

The authors present a relaxation of Evidential Deep Learning (EDL) methods that makes adjustments to the canonical model formulation and loss function.
EDL has recently gained popularity as an alternative family of uncertainty quantification methods that provides confidence estimates in a single forward pass by having neural networks parameterize distributions over distributions.
The authors argue that two details of the standard formulation of EDL are misguided:
Firstly, they argue that a scalar which gives weight to the prior compared to the evidence is often unnecessarily set to the number of classes in the classification problem, leading to sometimes unintuitive results.
To solve this, the authors propose to instead treat it as a task-specific hyperparameter.
Secondly, they make a case that optimizing prior networks using the expected MSE loss between the predicted probabilities and a one-hot target vector creates over-confident networks due to the variance term in the objective's decomposition.
Here, they propose to simply optimize the MSE loss directly, without the expectation.
They show in their experiments in a classical, few-shot setting for image classification and video action recognition datasets that their method, called R-EDL) able to better indicate the presence of OOD inputs and misclassified examples based on its uncertainty than their tested baselines, all while maintaining a high classification accuracy.

**Strengths:**

* The paper presents its ideas well: Despite the proposed changes being relatively minor in nature, it motivates them with illustrating examples and reiterates EDL's basis in subjective logic to motivate them.
* The presentation of the paper is polished and it is generally well written, albeit that earlier sections can feel slightly verbose at times.
* The paper compiles a number of relevant baselines that are used for comparison in the presented experiments.
* The authors test their approach in a classical, few-shot and noisy setting along with an ablation study.
* Experimental details for possible replications are documented well in the appendix.

**Weaknesses:**

* It would have felt more natural to me to include the Natural Posterior Network [1] instead of / besides the Posterior Network as a baseline, since it boasts better results than its predecessor.
* Regarding the creation of a new hyperparameter, I do appreciate the authors' ablation study in 5.5. However, the investigation, given that this step is one of the key contributions of the paper, feels a bit shallow. I would be interested to understand better how to interpret the best values found in the context of the authors' original argument and to distill any trends. I found Figure 1b) particularly puzzling, since lambda seems to have major effect on classification performance, but less so in terms of actual OOD detection (~2 points AUPR). A similar graph about misclassification detection, which seems to align better with the authors original arguments, is not given.
* Although the second contribution - effectively deprecating the variance term in the loss function - is supposed to help overconfidence, confidence is never directly evaluated. This might be due to the common conflation of the terms uncertainty and confidence, with the latter often referring to the actual probability assigned to predicted class - however works like the aforementioned Charpentier et al. [1] evaluate both the quality of uncertainty estimates obtained through different EDL metrics as well as the direct confidence of their model through the Brier score. The authors do try to assess confidence through measuring the AUPR for their misclassification task (including the MP value, i.e. the maximum probability across all classes), which I would argue is appropriate, but assesses confidence only in its ability to discriminate between correctly and incorrectly classified samples, not in its degree to correspond to the (in expectation) chance of classifying correctly, like in the instance of the expected calibration error.

[1] Charpentier, Bertrand, et al. "Natural Posterior Network: Deep Bayesian Predictive Uncertainty for Exponential Family Distributions." International Conference on Learning Representations. 2021.

**Questions:**

My questions are the follows:

* Was there any particular rational in not including Brier score / ECE in the evaluation?
* What was the reason for including the Posterior Network as a baseline, but not the Natural Posterior Network? (see weaknesses section)
* Even after consulting the appendix, it is unclear to me if and which regularization term has been used to trained the models (since e.g. Sensoy et al. [2] use an additional KL divergence term). Given the number of options for regularization (see e.g. [3] for an overview) it feels necessary to know which regularizer had been used. In any case, this part would deserve a short mention in the main text. If no regularizer had been used at all, I would find the good results for the OOD detection task surprising, since many other works lament that uncertainty on OOD behaves differently for EDL than desired.
* Judging from Table 3, I find it a bit strange that either fixing lambda = 1 or re-adding the variance term to the loss function seem to have only minor effects on the results of R-EDL compared to the EDL baseline. This implies to me that a) there might be an interaction between the two components that might go beyond the sum of its parts or that b) the EDL baseline might be tuned suboptimally compared to the proposed R-EDL model. I would be interested in the authors commenting on this aspect.

Other notes:
* The link to the appendix after the contributions in section 1 seems to be missing.

[2] Sensoy, Murat, Lance Kaplan, and Melih Kandemir. "Evidential deep learning to quantify classification uncertainty." Advances in neural information processing systems 31 (2018).

[3] Ulmer, Dennis, Christian Hardmeier, and Jes Frellsen. "Prior and posterior networks: A survey on evidential deep learning methods for uncertainty estimation." Transactions on Machine Learning Research (2023).

---

> ### Author Response · Authors · 2023-11-18
> **Response to Reviewer ErzN (Part 1/3)**
>
> We express our sincere gratitude for your insightful and constructive comments. We have meticulously revised the manuscript as suggested. Below, we first reiterate your comments, subsequently providing our detailed responses to each point.
>
> > Comment 1: The authors do try to assess confidence through measuring the AUPR for their misclassification task (including the MP value, i.e. the maximum probability across all classes), which I would argue is appropriate, but assesses confidence only in its ability to discriminate between correctly and incorrectly classified samples, not in its degree to correspond to the (in expectation) chance of classifying correctly, like in the instance of the expected calibration error. Was there any particular rational in not including Brier score / ECE in the evaluation?
>
> Thanks for the insightful comment. To address this concern, in the revised version, **we have enhanced the clarity and included results evaluated using the Brier score and Expected Calibration Error (ECE) in the Appendix as Table 8**. Furthermore, to elucidate our rationale for adopting the Area Under the Precision-Recall Curve (AUPR) as the evaluation metric, as suggested in reference [4], over the Brier score or ECE, we present our justification below:
>
> As delineated in Section 5.2, our stance is that a reliable uncertainty estimator should **(1)** assign higher uncertainties to misclassified samples than to correctly classified ones, and **(2)** assign higher uncertainties to out-of-distribution (OOD) samples than to in-distribution (ID) ones. In other words, **our primary criterion for assessing confidence estimation is the model's ability in differentiating between correctly classified and misclassified samples, as well as between ID and OOD samples based on the predicted confidence**. This is the fundamental reason for our selection of AUPR as the evaluation metric. The term "over-confidence," as used in our context, specifically pertains to the tendency of neural networks to assign excessively high confidence to misclassified or OOD samples, a perspective that deviates from some existing literature.
>
> We recognize that ECE is frequently employed to assess the degree of correspondence between the model's confidence and the true correctness likelihood, which is important for classification models in many applications. However, **a confidence distribution accompanied by a low ECE does not inherently ensure a clear distinction between correct and incorrect predictions**. For instance, in a balanced two-class dataset scenario, if a binary classifier categorizes all samples into a single class with a consistent confidence output of 50%, the ECE would be zero, yet this result lacks practical significance. Therefore, we posit that for the objectives of our study, AUPR provides a more pertinent evaluation. Furthermore, whereas ECE is focused on the absolute values of confidence, AUPR concentrates on the ranking order of confidence. We consider both metrics as direct evaluators of the quality of predicted confidence, each with its unique perspective and utility.
>
> | Method | ECE (15 bins)$\downarrow$ | Brier score$\downarrow$ | AUPR (confidence) | AUPR (OOD against SVHN)
> | :---: |:---: |:---: |:---:|:---:|
> | EDL | 20.08$\pm$1.77 | 40.68$\pm$2.39 |97.86$\pm$0.17|78.87$\pm$3.50|
> |I-EDL| 39.96$\pm$0.37 | 55.32$\pm$0.50 |98.72$\pm$0.12|83.26$\pm$2.44|
> |R-EDL| **3.48**$\pm$0.30 | **18.14**$\pm$0.51 |**98.98**$\pm$0.05|**85.00**$\pm$1.22|
>
> As shown in the above table, which summarizes results from five runs on CIFAR10, our proposed method outperforms EDL and I-EDL with a large margin when evaluated by the ECE and Brier score. It is mainly due to that we relax the rigid setting of fixing the prior weight parameter to the number of classes. With the original setting of $W$ (or $\lambda$), as illustrated in Section 3.2, it is difficult for previous EDL-related works to make predictions of high confidence, despite the fact that they actually output correct evidence distributions, thus resulting in excessively high ECE and Brier scores. With our modification, model's predicted confidence becomes much more aligned with the expected classification accuracy. It is noteworthy that, although previous EDL-related works paid little attention to the correspondence of confidence and expected accuracy, they are still effective in detecting misclassified and OOD samples, which also aligns with our primary objective.
>
> In summary, our stance is that **employing the AUPR scores for evaluation purposes aligns more closely with our objectives than using ECE or Brier score**. Additionally, for the sake of comprehensive analysis and transparency, the results evaluated using ECE and Brier score have been included in the Appendix as Table 8, serving as a reference point.
>
> [4] Deng, Danruo, et al. "Uncertainty estimation by fisher information-based evidential deep learning." ICML, 2023.

---

> ### Author Response · Authors · 2023-11-18
> **Response to Reviewer ErzN (Part 2/3)**
>
> > Comment 2: It would have felt more natural to me to include the Natural Posterior Network [1] instead of / besides the Posterior Network as a baseline, since it boasts better results than its predecessor. What was the reason for including the Posterior Network as a baseline, but not the Natural Posterior Network?
>
> We extend our gratitude for directing our attention to the important work "Natural Posterior Network" (NatPN) [1]. The primary reason for the initial omission of [1] from our analysis was our reliance on the baseline selection in reference [4] from ICML 2023. Given that [1] did not conduct experiments using the CIFAR10 dataset with a VGG16 backbone, we reproduced their method and generated the following results, which were assessed using the same metrics as those presented in Table 1:
>
> | Method | Classification Accuracy | Confidence Estimation | OOD Detection (SVHN) | OOD Detection (CIFAR100)|
> | :---: | :---: | :---: | :---: | :---: |
> | PostN | 84.85$\pm$0.0 | 97.76$\pm$0.0 | 80.21$\pm$0.2 | 81.96$\pm$0.8 |
> | NatPN | 84.72$\pm$0.0 | 97.81$\pm$0.0 | 80.14$\pm$0.3 | 81.80$\pm$1.0 |
> | R-EDL | **90.09**$\pm$0.30 | **98.98**$\pm$0.05 | **85.00**$\pm$1.22 | **87.72**$\pm$0.31 |
>
> It is noteworthy that the performance of NatPN does NOT surpass that of its predecessor, which aligns with the assertions made by the authors in [1]. They state that a single NatPN achieves accuracy and calibration performance **comparable to** PostNet. The principal advantage of NatPN lies in its efficiency, as it utilizes a single normalizing flow, thereby offering a significant speed advantage over PostNet, whose computational requirements scale linearly with the number of classes due to its evaluation of a separate normalizing flow for each class.
>
> [4] Deng, Danruo, et al. "Uncertainty estimation by fisher information-based evidential deep learning." ICML, 2023.
>
> > Comment 3: Regarding the creation of a new hyperparameter, I do appreciate the authors' ablation study in 5.5. However, the investigation, given that this step is one of the key contributions of the paper, feels a bit shallow. I would be interested to understand better how to interpret the best values found in the context of the authors' original argument and to distill any trends.
>
> We appreciate your insightful comment.
>
> To interpret the best values, in our discussion in Section 3.2, we elaborate on the significance of the parameter $W$ (or $\lambda$) in influencing the projected probability $P_X(x)$, specifically in terms of the magnitude and proportion of evidence. Theoretically, for any specific case, there should be an optimal value of $W$ that effectively balances the trade-off between utilizing the proportion and magnitude of evidence. This optimal value is what we refer to as the "best value." As evidenced in Figure 1(b), this best value is not a singular point but rather spans across a suitable continuous range.
>
> The trend analysis follows a similar logic. At the point where $W$ (or $\lambda$) is reduced to zero, the model's predictive probability is entirely dependent on the proportion of evidence. Conversely, as $W$ (or $\lambda$) increases substantially, the model progressively prioritizes the magnitude of evidence over its proportion. This trend is visually represented in Figure 1(b), where it is clear that an over-reliance on either magnitude or proportion leads to suboptimal performance.
>
> We acknowledge that our current interpretation does not comprehensively elucidate the intricate mechanisms that determine the optimal value of this parameter. Indeed, deciphering this aspect poses a significant challenge, as the output of modern neural networks is shaped by a plethora of complex factors, including dataset characteristics, architectural choices of the backbone, etc., rather than being solely dependent on the number of classes or any isolated factor. As we have posited in the Conclusion (Section 6) of our paper, *the underlying mechanism dictating its optimal value is a topic worth further investigation*.
>
> > Comment 4: I found Figure 1b) particularly puzzling, since lambda seems to have major effect on classification performance, but less so in terms of actual OOD detection (~2 points AUPR).
>
> We are grateful for the valuable comment.
>
> In our view, it is reasonable that the value of $\lambda$ impacts the accuracy of classification. As the value of $\lambda$ increases, the neural network is required to generate evidence for the target class with greater absolute values to attribute a high level of confidence to its predictions, which can escalate the propensity for overfitting. We conjecture that this phenomenon may contribute to the comparatively lower classification accuracy (83.55%) observed for EDL as presented in Table 1.

---

> ### Author Response · Authors · 2023-11-18
> **Response to Reviewer ErzN (Part 3/3)**
>
> > Comment 5: A similar graph about misclassification detection, which seems to align better with the authors original arguments, is not given.
>
> We appreciate your suggestion. **Now we have included this graph in the appendix of the revised version.** As shown in Figure 2(b) of the revised version, this graph displays a trend similar to the existing two parameter analysis graphs, characterized by an initial increase followed by a subsequent decline.
>
> > Comment 6: Even after consulting the appendix, it is unclear to me **if and which regularization term has been used to trained the models** (since e.g. Sensoy et al. [2] use an additional KL divergence term). Given the number of options for regularization (see e.g. [3] for an overview) it feels necessary to know which regularizer had been used. In any case, this part would deserve a short mention in the main text. If no regularizer had been used at all, I would find the good results for the OOD detection task surprising, since many other works lament that uncertainty on OOD behaves differently for EDL than desired.
>
> We express our gratitude for your valuable suggestion.
>
> We indeed followed Sensoy et al. [2] to use the additional KL divergence term, whose specific form is introduced in Appendix A.2, and its implementation is available in our provided source codes. Since our main baselines, EDL and I-EDL, both adopt this term, we do not pay much attention to it.
>
> **In the revised version, we have clarified this aspect explicitly in the main text.** Thanks again for the suggestion.
>
> > Comment 7: Judging from Table 3, I find it a bit strange that either fixing lambda = 1 or re-adding the variance term to the loss function seem to have only minor effects on the results of R-EDL compared to the EDL baseline. This implies to me that a) there might be an interaction between the two components that might go beyond the sum of its parts or that b) the EDL baseline might be tuned suboptimally compared to the proposed R-EDL model.
>
> Thank you for your perceptive remark. We intend to address this issue from two perspectives.
>
> Firstly, the hypothesis suggesting that *"there might be an interaction between the two components that might go beyond the sum of its parts"* does not appear to be substantiated by our findings. As evidenced in Table 3, particularly in the column pertaining to OOD detection against SVHN, we observe that R-EDL with a fixed lambda value of 1 yields an improvement of ~4%. When the variance term is reintroduced in R-EDL, the improvement is ~5%, and with the complete R-EDL model, the enhancement is ~6%. This suggests that the combined effect of the two components (6%) does not exceed their cumulative impact (4% + 5%), which is a relatively common occurrence in the field of deep learning. Should there be any aspect of your comment that we have misinterpreted, we are open to and encourage further discussion on this matter.
>
> Secondly, regarding the EDL baseline, we utilized the [official code](https://github.com/danruod/IEDL) from I-EDL [4]. Our results align with those reported in [4] and are reproducible, confirming the reliability of our experimental setup and findings.
>
> [4] Deng, Danruo, et al. "Uncertainty estimation by fisher information-based evidential deep learning." ICML, 2023.
>
> > Comment 8: The link to the appendix after the contributions in section 1 seems to be missing.
>
> Thanks for the reminder! It has been fixed.
>
> **Finally, we extend our sincerest thanks for all your time and consideration!**

---

> ### Comment · Reviewer_ErzN · 2023-11-20
> **Response to author clarifications**
>
> I would like to thank the authors for their detailed feedback and effort to address all of my points raised. While I do still think that some of the results are surprising or warrant further investigation, I believe this should not be a hindrance to having this manuscript published. I also do not think that the paper should be disqualified for seemingly simple or incremental improvements; changes appear well-motivated and supported by empirical evidence. The updated draft also adds context and additional results that were missing for me from the original version. I updated my scores accordingly.

---

### Author Response · Authors · 2023-11-18
**Responses to reviewers and a new version of our work have been uploaded.**

We thank all the reviewers for their thorough reading of our work and the high-quality feedback they provided. Their comments have been immensely beneficial in enhancing the quality of our manuscript and deepening our own understanding of this field. We have uploaded responses for each reviewer along with a revised version of the paper, which we believe addresses all the issues raised.

Paper revisions relating to the feedback from each reviewer have been meticulously detailed in the responses. We extend our sincerest thanks to all reviewers for their time and consideration!

---

### Author Response · Authors · 2023-11-23
**Summary of the rebuttal. We are looking forward to your reply!**

Dear reviewers,

We would like to express our sincere gratitude for the valuable feedback you previously provided. We have revised our manuscript in accordance with your suggestions and believe that these modifications enhance the quality of our work. Allow me to summarize the significant alterations made in both the rebuttal and the revised manuscript:

1. **Additional results evaluated by more metrics**: Included additional results evaluated by ECE and Brier score in Table 8 of the appendix.
2. **Discussion about metric choices**: Added a discussion about why we chose AUPR instead of ECE as the main metric to the appendix D.1.
3. **More hyper-parameter analysis**: Included a graph of the hyper-parameter analysis about confidence estimation as Figure 2 in the appendix.
4. **Comparison with temperature scaling**: Included the performance of the temperature scaling method for a more comprehensive comparison in Table 8.
5. **More related work**: Added a discussion about "Evidential Turing Processes, ICLR 2022" to Section 4.
6. **More uncertainty density plots**: Added uncertainty density plots (Figure 7-10) for ID (MNIST) and OOD (FMNIST, KMNIST) to the appendix.
7. **Revised Writing and Further Explanations**: Enhanced the clarity of concepts such as belief mass and uncertainty mass, added explanations to the different shapes of uncertainty density plots, and corrected typos.

I understand that reviewers may be extremely busy. However, **with the discussion deadline less than 8 hours away**, we would greatly appreciate your prompt response. Should you have any further suggestions or comments, we are more than willing to consider them and make the necessary adjustments.

Thank you again for your valuable time and expertise throughout this process. We eagerly await your reply and hope to complete all required revisions before the deadline.

Warm regards,

Authors of submission 597

---

### Meta-Review · Area_Chair_Lvyu · 2023-12-06

**Metareview:**

The authors propose a relaxation to the Evidential Deep Learning (EDL) framework, addressing two critical issues in the standard formulation by treating the prior weight as a hyperparameter and directly optimising the expected value of the constructed Dirichlet in the objective function. The experiments demonstrate that this new method, R-EDL, has SOTA performance in confidence estimation and OOD detecting both in the standard classification setting and in a few-shot setting while maintaining high classification accuracy.

**Justification For Why Not Higher Score:**

While all reviewers recommend accepting the paper, two note that the proposed methodology is somewhat incremental.

**Justification For Why Not Lower Score:**

The paper proposed a new methodology that is backed by solid empirical results. All reviewers recommend accepting this paper, and the authors have carefully addressed most of the reviewers' concerns, which three of the reviews also have acknowledged.

---

### Decision · Program_Chairs · 2024-01-16

Accept (spotlight)